# Multi-objective optimization of dual resource integrated scheduling problem of production equipment and RGVs considering conflict-free routing

Qinglei Zhang[1,2], Jing Hu[2]*, Zhen Liu[2], Jianguo Duan[1,2]

**1** China Institute of FTZ Supply Chain, Shanghai Maritime University, Shanghai, China, **2** Institute of Logistics Science and Engineering, Shanghai Maritime University, Shanghai, China

* janette10@163.com

**Data Availability Statement:** All relevant data are within the manuscript and its Supporting Information files.

## Abstract

In flexible job shop scheduling problem (FJSP), the collision of bidirectional rail guided vehicles (RGVs) directly affects RGVs scheduling, and it is closely coupled with the allocation of production equipment, which directly affects the production efficiency. In this problem, taking minimizing the maximum completion time of RGVs and minimizing the maximum completion time of products as multi-objectives a dual-resource integrated scheduling model of production equipment and RGVs considering conflict-free routing problem (CFRP) is proposed. To solve the model, a multi-objective improved discrete grey wolf optimizer (MOID-GWO) is designed. Further, the performance of popular multi-objective evolutionary algorithms (MOEAs) such as NSGA-II, SPEA2 and MOPSO are selected for comparative test. The results show that, among 42 instances of different scales designed, 37, 34 and 28 instances in MOID-GWO are superior to the comparison algorithms in metrics of generational distance (GD), inverted GD (IGD) and Spread, respectively. Moreover, in metric of Convergence and Diversity (CD), the Pareto frontier (PF) obtained by MOID-GWO is closer to the optimal solution. Finally, taking the production process of a construction machinery equipment component as an example, the validity and feasibility of the model and algorithm are verified.

## 1. Introduction

With the promotion of Industry 4.0, to improve production efficiency under the demand of batch and individualized production, two or more single-load bi-directional rail guided vehicles (RGVs) are often used to transport large-size and large-tonnage products and components on the single track [1]. At present, the existence of rail occupying and bi-directional conflict between RGVS leads to the occurrence of delayed loading and unloading problems. Although the flexible job-shop scheduling problem (FJSP) [2] research is extensive, most of them do not consider the transportation time, or treat the transportation time between workstations as a constant value, which is a weak guide for the actual production units where RGVs undertake logistics and transport.

**Funding:** The author(s) received no specific funding for this work.

**Competing interests:** The authors have declared that no competing interests exist.

Further, the scheduling of RGVs directly affects the change of production equipment scheduling. Although some scholars have focused on RGV scheduling optimization, such as considering the conditions of energy constraint [3], speed constraint [4] and conflict constraint [5]. However, adding RGV conflict constraints to FJSP will significantly increase the complexity of the problem. Therefore, there is little research on dual resource and multi-objective integrated scheduling of RGV and production equipment. To solve the problem of delayed loading and unloading in the manufacturing process caused by route conflict between RGVs, and improve the production efficiency of integrated operation between RGV and production equipment, the multi-objective optimization of dual resource integrated scheduling problem of production equipment and RGVs considering conflict-free routing (DRISP-PERCFR) is researched, and a non-linear integer programming model is developed. Grey wolf optimizer (GWO) has good performance in both convergence speed and solving accuracy, but the original algorithm is mainly used to solve continuous and single-objective optimization problems [6]. A new multi-objective improved discrete GWO (MOID-GWO) is designed to solve the model and tested in different instances. The main innovations are as follows:

1. The integrated scheduling model of DRISP-PERCFR is established, and a new RGVs conflict avoidance solution is proposed in the coupling relationship between transportation and production The route conflicts of RGVs are described more detail, and the loading/ unloading occupancy conflict, waiting occupancy conflict and transportation conflict models are constructed and solved respectively.

2. MOID-GWO is proposed to solve the above problems. To improve the quality and diversity of the initial population, The reducing job process time, transportation time, loading and unloading time strategy (RPTL), and the improved auction protocol heuristic strategy (IAPH) are developed to solve the dependence of GWO on the quality and diversity of initial solutions.

3. In MOID-GWO, an improved discrete combinational optimization search operator is designed to update the individual location of grey wolves, and the local search capability is strengthened by designing a neighborhood structure.

4. By constructing different scale test examples, the performance index of Pareto optimal solution is compared with NSGA-II, strength Pareto evolutionary algorithm 2 (SPEA2), and multi-objective particle swarm optimization algorithm (MOPSO).

5. The effectiveness of DRISP-PERCFR's model and MOID-GWO is verified by a practical case of a welding manufacturing enterprise.

The rest of the paper is organized as follows: Section 2 provides a literature review related to the problem. In Section 3, related problems are explained and mathematical models are established. In Section 4, MOID-GWO is designed. In Section 5, experimental research is carried out. Anda practical case study is analyzed in Section 6. Conclusions and future work are introduced in Section 7.

## 2. Literature review

### 2.1. Integrated scheduling problem of transportation and production equipment

There is little research on the integrated scheduling of RGV and production equipment, but the automatic guided vehicle (AGV) and RGV have similar working environments and tasks. Therefore, it is of certain reference significance to analyze the integrated scheduling problem of AGV and production equipment.

Aiming at minimizing production cycle time and total delay, Rahman [7] proposed an integrated scheduling model and adopted a phased strategy to achieve the balance between the robot assembly line and AGV material handling Chaudhry [8] proposed an Excel-based method to deal with the integrated scheduling problem of machines and AGV. A proprietary genetic algorithm Evolver was used to allocate the machine and allocate the AGV autonomously. To some extent, this model is also phased. Yao [9] proposed an intelligent AGV management system that maximizes accurate delivery performance and minimizes AGV energy consumption under disturbance by establishing a mixed integer nonlinear programming model supported by IoT production data. Li [10] proposed a genetic tabu search algorithm to solve the minimum maximum completion time of integrated scheduling of production and transportation, and proved its effectiveness through the results. Yan [11] analyzed the coupling relationship between the transportation and the processing stage and established the scheduling model under the limited transportation conditions given the constraints of the limited transportation.

Since AGVs travel in one direction, most of the above studies ignore the conflict-free routing problem (CFRP). However, RGVs usually run along a fixed track and can travel in both directions, so route conflict is inevitable. Therefore, it is very necessary to consider the collision problem in the RGV scheduling problem.

## 2.2. Conflict-free routing problem (CFRP)

To obtain a more realistic maximum completion time result, CFRP has been paid more and more attention.

Li [12] introduced constraints of time window (TW) and conflict resolution mechanisms to obtain cutting job assignments and scheduling with CFRP. Kaboudani [13] made a comparative study on the vehicle route problem (VRP) with the lowest transportation cost in a distribution network with a cross-docking center. Nishid [14] studied the solution of CFRP for just-in-time delivery of AGVs, which solves the AGV scheduling with minimum total penalties for early and late arrival of dynamic tasks. Ma [5] carried out dual RGV scheduling considering both bilateral loading and unloading constraints and collision avoidance constraints and established a mathematical model to minimize the maximum completion time. Chang [15] combined an improved banker algorithm (IBA) with TW to propose an RGV transport blocking and deadlocking control strategy, which ensures RGV operation security and improves RGV operation efficiency. Goli [16] used an improved moth flame meta-heuristic algorithm to solve CFRP, and the vehicle operating cost in the model is considered. However, the above studies mainly focus on the traveling salesman problem (TSP) and CFRP, and largely ignore the problem of shop scheduling.

## 2.3. Integrated scheduling problem of transportation equipment with CFRP and production equipment

Cao [17] adopts the two-layer planning method to solve the problem of integrated scheduling between container inbound and outbound tasks and AGV bidirectional CFRP. Generally, the common solution strategy for integrated scheduling problems of AGVs with CFRP and production equipment is to divide the whole problem into two stages independently, that is, firstly determining the optimal AGVs and production equipment scheduling solution, and then optimizing VRP. It cannot guarantee that the optimal scheduling solution matches the optimal solution of path planning [18]. Although Lyu [19] combined the completion time and AGV running time considering CFRP, the study regarded CFRP as equivalent or as punishment.

**Table 1. Comparison of popular evolutionary algorithms.**

| Evolutionary algorithms | Global search capability | Rate of convergence | Simplicity and ease of implementation | Applicability of complex constraints |
|---|---|---|---|---|
| GWO | Strong | Fast | Facile | Applicative |
| Simulated Annealing (SA) [25] | Moderate | Slow | Facile | Hard |
| Hill-climbing | Weak | Slow | Facile | Hard |
| GA | Moderate | Moderate | Moderate | Moderate |
| Artificial Bee Colony (ABC) [26] | Strong | Moderate | Moderate | Moderate |
| PSO | Moderate | Moderate | Moderate | Moderate |
| Differential Evolution (DE) | Moderate | Moderate | Moderate | Moderate |

## 2.4. Evolutionary algorithms

The integrated scheduling problem of transportation equipment and production equipment is a typical NP-hard problem. For large-scale problems, traditional algorithms, such as the dynamic programming method and branch and bound method, can not achieve the speed and accuracy of small-scale scheduling optimization problems. This leads researchers to study evolutionary algorithms (EA), like penguin search optimization algorithm (PeSOA) [20], GA [21], cuckoo search (CS) [22] and particle swarm optimization (PSO) [23]. The current widely used evolutionary algorithms (EA) [24] are shown in Table 1, which comprehensively considers the global search ability, Rate of convergence, simplicity and ease of implementation, applicability of complex constraint. Although PSO, EA, GA and ABC perform well in Simplicity and ease of implementation and complex constraint applicability, many parameter adjustments are involved, resulting in unstable results.

Compared with other algorithms, grey wolf optimizer (GWO) has better performance in terms of solution quality and stability, and has been widely used [27]. The original GWO flow can be found in [28].

To summarize, the following deficiencies and challenges can be found:

1. Few researches on the integrated scheduling problem of RGV and production equipment with bidirectional transportation and CFRP.

2. The integrated scheduling problem of transportation equipment and production equipment is mostly studied in stages. CFRP of transportation equipment is not considered, or CFRP is equivalent. The single objective of minimizing the maximum completion time is studied, and the transportation time should be reduced as much as possible when solving CFRP.

3. GWO is mostly used to solve continuous function problems, which has certain challenges for solving discrete optimization problems.

Based on the above shortcomings, aiming at the minimization of the maximum transportation time and completion time, DRISP-PERCFR is studied and MOID-GWO is proposed to solve the problem.

## 3. Problem description and mathematical model

### 3.1. Problem description

DRISP-PERCFR can be described as follows:

As shown in Fig 1, there are 2 identical, single-load, bi-directional transport RGVs sharing a single track used for loading/unloading and transporting workpieces in the operating system.

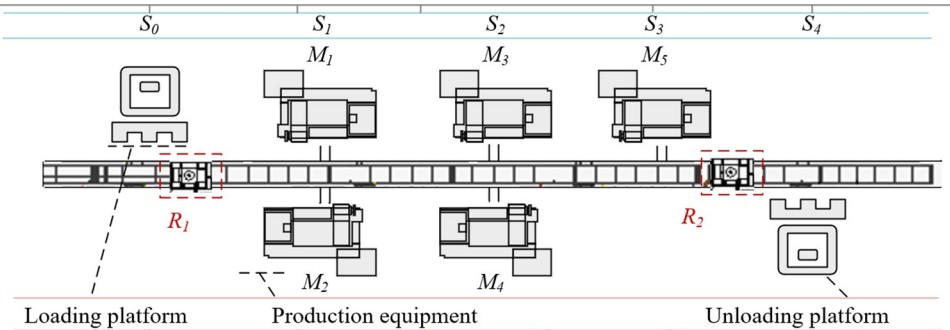

**Fig 1. An example of DRISP-PERCFR.**

The *m* production machines, subsequently called machines, are located on both sides of the track. Each process can be processed on several different machines.

The RGV is a daughter car and occupies track station *S* when performing loading and unloading tasks. Before scheduling begins, $R_1$ and $R_2$ of RGV are located in the leftmost loading platform $S_0$, and the rightmost unloading platform $S_4$, respectively [29]. The focus here needs to be on two problems: The first is the machine scheduling, including the selection of the machine and the processing sequence of workpieces [30]. The second is the RGVs scheduling, including the selection of RGVs and the optimal pick-up and delivery routes considering CFRP.

The transport task of RGV consists of unloaded transport, loaded transport, loading/unloading, as shown in Fig 2. During the loading/unloading task, the RGV needs to stay on track. Irrational scheduling planning leads to route conflicts, which can lead to production stoppages or major accidents.

Examples of assignments considered and not considered CFRP are shown in Fig 3. When time $t = 0$, $R_1$ is at $S_0$, and $R_2$ is at $S_5$. The RGVs transmit assigned requests in sequence and return to their respective starting positions when the task is completed. When CFRP is not considered, as shown in Fig 3(A), the RGVs perform tasks according to pre-planned routes. The collision occurs at points A and B at times $t = 68$min and $t = 82$min, respectively. When considering CFRP, $R_2$ replans the route to avoid the collision, as shown in Fig 3(B).

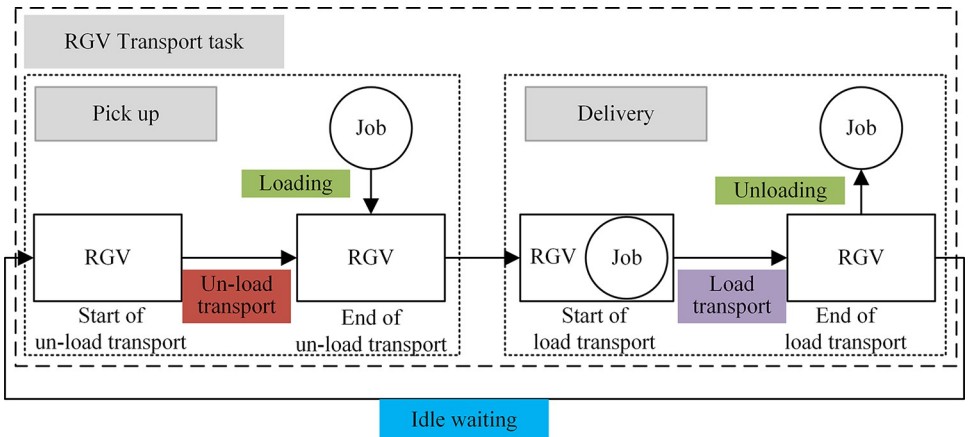

**Fig 2. Schematic diagram of RGV transport tasks.**

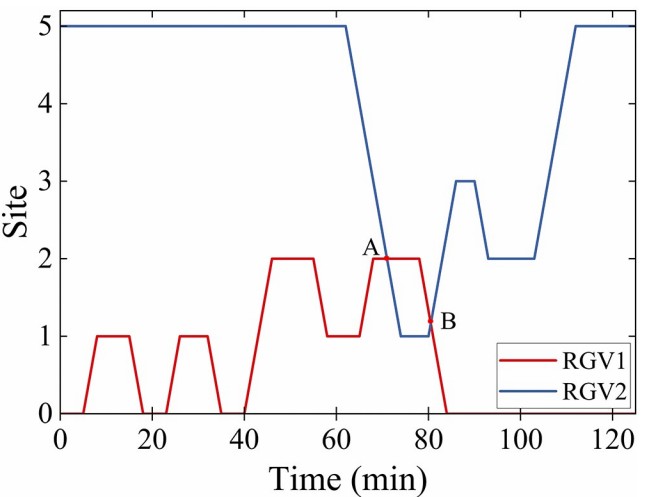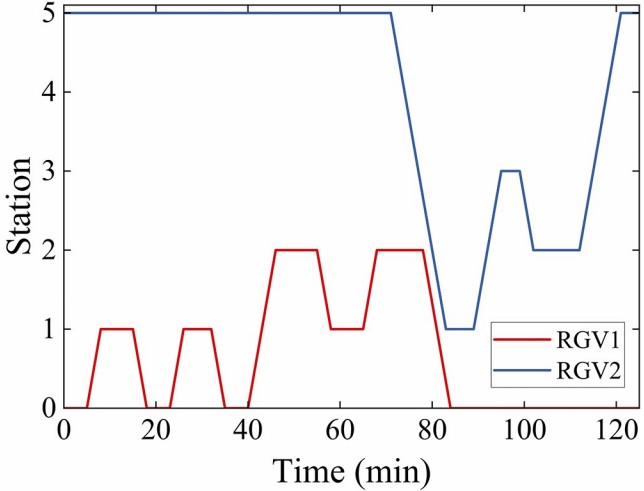

**Fig 3.** The routes of RGVs: (a) with CFRP; (b) without CFRP.

A directed graph $G = \{V,E\}$ is used to express the transport task path. $V = \{V_1,V_2\}$ is the vertex represented by the stop stations in the route. $V_v^i$ is the $i_{\text{th}}$ vertex of $V_v$. $E$ is the set of edges of the vertices. $tSV_v^i$ and $tCV_v^i$ represent the arrival and departure moments, respectively. $tUV_v^i$ represents the completed moment of the loading and unloading tasks of $V_v^i$. $SV_v^{ij}$ represents all stations transported from $V_v^i$ to $V_v^j$, including $V_v^i$ and $V_v^j$ $EV_e$ represents the type of conflict.

The following is a definition of conflict:

**Def. 1**: If both Eqs 1 and 2 are satisfied, then it is a loading/unloading occupancy conflict, named as $EV_1$.

$$V_{v'}^{i'} \in SV_v^{i(i+1)}$$
$$[tCV_v^i, tSV_v^{i(i+1)}] \cap [tSV_{v'}^{i'}, tCV_{v'}^{i'}] \neq \emptyset$$
<div align="right">Eq1</div>

$$tUV_v^i > tSV_v^i$$
<div align="right">Eq2</div>

**Def. 2**: If both Eqs 1 and 3 are satisfied, then it is a waiting occupancy conflict, named as $EV_2$.

$$tCV_v^i > tUV_v^i$$
<div align="right">Eq3</div>

**Def. 3**: If Eq 4 is satisfied, then it is a transport conflict, named as $EV_3$.

$$SV_v^{i(i+1)} \cap SV_{v'}^{i'(i'+1)} \neq \emptyset$$
$$[tCV_v^i, tSV_v^{i(i+1)}] \cap [tCV_{v'}^{i'}, tSV_{v'}^{i'(i'+1)}] \neq \emptyset$$
<div align="right">Eq4</div>

### 3.2. Mathematical modelling

The assumptions of DRISP-PERCFR are as follows:

1. RGVs are the same. The transport speed is uniform and the same, which does not take into account the acceleration at the starting and the deceleration at the stopping.

2. RGVs can transport all workpieces and only one workpiece can be transported at the same time.

3. RGVs operate without incident and transport tasks are not interrupted.

4. RGVs can reach any position on the track, and operation is bi-directional.

5. RGV with intelligent control, can receive and send command signals in a short, negligible amount of time.

6. RGVs are located at the left and right ends of the track respectively at the initial moment.

7. When completing the transport task, RGVs stay at the station.

8. Machines are symmetrically distributed along the track with fixed and known spacing.

9. Machines are idle at the initial moment.

10. Machines operate normally without downtime and fault.

11. Symmetric machines can only be accessed by a single RGV.

12. Machines do not provide a workpiece buffer zone. In other words, production can only continue after the workpiece has been delivered.

13. Only one workpiece can be processed by each machine at one time, and each workpiece can only be processed by one machine at the same time.

14. The locations of the loading and unloading platforms are distributed at both ends of the track, and the spacing between the nearest machine is equal to the spacing between machines.

The defined variables of interest are as follows:

| | |
|---|---|
| $N_i$ | Workpieces, $i \in \{1,2,\ldots,n\}$ |
| $M_k$ | Machines, $k \in \{0\} \cup \{1, 2, \ldots, m\} \cup \{m + 1\}$. $\{0\}$ and $\{m+1\}$ represent the equivalent machine number for the loading and unloading platform, respectively. |
| $R_v$ | RGV, $v \in \{1,2\}$ |
| $p_i$ | Number of processes for $N_i$ |
| $O_{ij}$ | The jth process of $N_i$, $j \in \{1, 2, \ldots, p_i\} \cup \{p_i + 1\}$. $\{p_i+1\}$ represents the virtual transport process to the unloading platform. |
| $S'$ | Station corresponding to the machine. The station where $M_k$ is located is $\lceil M_k/2 \rceil$, where $\lceil \rceil$ represents rounding upwards. |
| $S$ | Station. $S = \{0\} \cup \{S'\} \cup \{\lceil m/2 \rceil + 1\}$. $\{0\}$ and $\{\lceil m/2 \rceil+1\}$ represent the loading and unloading platform respectively. |
| $\theta$ | Minimum safe distance for two RGVs to avoid collision, generally selected as a multiple of the machine pitch |
| $l_r$ | Spacing between neighboring stations. |
| $t$ | Moment, initial moment $t = 0$ |
| $P_{Rvt}$ | Position of $Rv$ at time $t$ |
| $P_s$ | Position of the station in the track. $P_s = s \times l_r, s \in S$ |
| $T_{ijv}$ | Transportation task by $R_v$ for $O_{ij}$ |
| $M_{ijk}$ | Production task by $M_k$ for $O_{ij}$ |
| $t^s_{Tijv}$ | Starting moment of $T_{ijv}$ |
| $t^f_{Tijv}$ | Ending moment of $T_{ijv}$ |
| $tS_{ijk}$ | Process starting moment of $O_{ij}$ in $M_k$ |
| $tC_{ijk}$ | Process ending moment of $O_{ij}$ in $M_k$ |
| $tSlT^n_{ijv}$ | Starting moment of the unloaded transport phase of $R_v$ for $T_{ijv}$ |
| $tElT^n_{ijv}$ | Ending moment of the unloaded transport phase of $R_v$ for $T_{ijv}$ |

*(Continued)*

**Eq 4.** (Continued)

| | |
|---|---|
| $tSlT_{ijv}^f$ | Starting moment of the loaded transport phase of $R_v$ for $T_{ijv}$ |
| $tElT_{ijv}^f$ | Ending moment of the loaded transport phase of $R_v$ for $T_{ijv}$ |
| $T_{Tijv}^{ts}$ | Mandate time for $T_{ijv}$ |
| $T_{Mijk}^{ms}$ | Processing time for $M_{ijk}$ |
| $T_{ijv}^{WT}$ | Waiting time of $R_v$ for $T_{ijv}$ |
| $T_{ijkv}^u$ | Time of $R_v$ responsible for $T_{ijv}$ to unload workpiece to $M_k$ |
| $T_{ijkv}^{u'}$ | Time of $R_v$ responsible for $T_{ijv}$ to load workpiece from $M_k$ to $R_v$ |
| $T_{Tijv}^n$ | Unloaded transport time of $R_v$ responsible for $T_{ijv}$ |
| $T_{Tijv}^f$ | Loaded transport time of $R_v$ responsible for $T_{ijv}$ |
| $G$ | Directed graph of routes $G = \{V,E\}$ |
| $V$ | $V = \{V_1,V_2\}$ |
| $V_v^i$ | The $i_{\text{th}}$ vertex of $V_v$ |
| $E$ | Edges of the vertices |
| $EV_e$ | Type of conflict |
| $B_v^{ij}$ | Security station added between original $V_v^i$ and $V_v^j$ to avoid conflicts |
| $tSV_v^i$ | Arrival moment of $V_v^i$ |
| $tCV_v^i$ | Departure moment of $V_v^i$ |
| $tUV_v^i$ | Loading or unloading completed moment of $tUV_v^i$ |
| $SV_v^{ij}$ | All stations transported from $V_v^i$ to $V_v^j$, including $V_v^i$ and $V_v^j$ |
| $M$ | An exceptionally large number |
| $v_R$ | Speed of the RGV during transport |
| $\alpha_{ijk}$ | $\alpha_{ijk}$ is 1 if $O_{ij}$ is processed in $M_k$ and 0 otherwise |
| $\beta_{ijv}$ | $\beta_{ijv}$ is 1 if $O_{ij}$ is processed in $R_v$ and 0 otherwise |
| $x_{O_{ij}O_{pq}k}$ | $x_{O_{ij}O_{pq}k}$ is 1 if $O_{ij}$ precedes $O_{pq}$ by 1 in order in $M_k$, and 0 otherwise |
| $y_{O_{ij}O_{pq}v}$ | $y_{O_{ij}O_{pq}v}$ is 1 if $O_{ij}$ precedes $O_{pq}$ by 1 in order in $R_v$, and 0 otherwise |
| $z_{svt}$ | $z_{svt}$ is 1 if $R_v$ is at station $s$ at $t$, and 0 otherwise |

The constraints are as follows:

$$f_1 = \min(\max_{i=1}^n C_i) \qquad\qquad \text{Eq5}$$

$$C_i = \sum_{j=0}^{p_i} \sum_{v=1}^2 \sum_k^m (T_{ijv}^{WT}\beta_{ijk} + T_{Tijv}^{ts}\beta_{ijk} + T_{Mijk}^{ms}\alpha_{ijv}) \qquad\qquad \text{Eq6}$$

$$T_{pqv}^{WT} = \sum_{i=1}^n \sum_{j=1}^{p_i} \sum_{p=1}^n \sum_{q=1}^{p_p} (t_{Tpqv}^s - t_{Tijv}^f)y_{O_{ij}O_{pq}v} \qquad\qquad \text{Eq7}$$

$$f_2 = \min(\max_{v=1}^2 VET_v) \qquad\qquad \text{Eq8}$$

$$VET_v = \sum_{i=1}^n \sum_{j=1}^{p_i} (T_{Tijv}^{ts}\beta_{ijv}) \qquad\qquad \text{Eq9}$$

$$\sum_{k=1}^{m} \alpha_{ijk} = 1, \forall i \in \{1, 2, \ldots, n\}, \forall j \in \{1, 2, \ldots, p_i\} \tag{Eq10}$$

$$\sum_{v=1}^{2} x_{ijv} = 1, \forall i \in \{1, 2, \ldots, n\}, \forall j \in \{1, 2, \ldots, p_i\} \tag{Eq11}$$

$$tS_{pqk} + L(1 - x_{O_{pq}O_{ij}k}) \geq tC_{ijk} \tag{Eq12}$$

$$tC_{ijk} = tS_{ijk} + T_{Mijk}^{m} \tag{Eq13}$$

$$t_{Tpqv}^{s} + L(1 - y_{O_{ij}O_{pq}v}) \geq t_{Tijv}^{s} \tag{Eq14}$$

$$t_{Tpqv}^{s} \geq t_{Tijv}^{f} y_{O_{ij}O_{pq}v} \\ t_{Tpqv}^{s} \geq tC_{p(q-1)k} \tag{Eq15}$$

$$t_{Tijv}^{s} = tSlT_{ijv}^{n} \tag{Eq16}$$

$$tElT_{ijv}^{n} = tSlT_{ijv}^{n} + T_{Tijv}^{n} \tag{Eq17}$$

$$tSlT_{ijv}^{f} = tElT_{ijv}^{n} + T_{ijkv}^{u'} \tag{Eq18}$$

$$tElT_{ijv}^{f} = tSlT_{ijv}^{f} + T_{Tijv}^{f} \tag{Eq19}$$

$$t_{Tijv}^{f} = tElT_{ijv}^{f} + T_{ijkv}^{u} \tag{Eq20}$$

$$tS_{ijk} \geq t_{Tijv}^{f} \\ tS_{ijk} \geq t_{Tijv}^{f} x_{O_{ij}O_{pq}k} \\ tS_{ijk} \geq tC_{i(j-1)k'} \tag{Eq21}$$

$$\sum_{s=0}^{S} \sum_{v=1}^{2} z_{svt} \leq 1 \tag{Eq22}$$

$$P_{R_1 0} = S_0, R_1 \in R \\ P_{R_2 0} = S_{\lceil m/2 \rceil + 1}, R_2 \in R \tag{Eq23}$$

$$P_{R_1 t} \leq P_{R_2 t} - \theta \tag{Eq24}$$

$$\alpha_{ijv} \in \{0, 1\}$$
$$\beta_{ijk} \in \{0, 1\}$$
$$\delta_{svt} \in \{0, 1\}$$
$$x_{O_{ij}O_{pq}k} \in \{0, 1\}$$    Eq25
$$y_{O_{ij}O_{pq}v} \in \{0, 1\}$$
$$i, p \in \{1, 2, \ldots n\}, j \in \{1, 2, \ldots p_i\}, q \in \{1, 2, \ldots p_p\}, v \in \{1, 2\}$$

Eqs 5–7 represent the objective function of the minimum maximum completion time. Eq 6 indicates that the completion time of the workpiece includes the total waiting time before transport tasks, the total time of transport tasks and the total processing time. Eq 7 indicates that the waiting time includes the relationship between the completed moment of the previous task and the start moment of this task.

Eqs 8 and 9 represent the objective function of minimum maximum transport task time.

Eq 10 represents that only one machine can be selected for each process. Eq 11 represents that only one RGV can be selected for the transport task. Eq 12 indicates that only one machine can process one workpiece at the same time. Eq 13 indicates that the workpiece cannot be interrupted once machining has begun. Eq 14 indicates that an RGV can only transport one task at the same moment. Eq 15 indicates that the RGV can start this task not earlier than the completed moment of the previous task and process of the workpiece. Eqs 16–20 indicate the time continuity of the transport task. Eq 16 indicates that the start moment of the task is equal to the start moment of unloaded transport. Eq 17 indicates that the unloaded transport end moment is equal to the sum of the unloaded transport start moment and unloaded transport time. Eq 18 indicates that the loaded transport start moment is equal to the sum of the unloaded transport end moment and the loading time. Eq 19 indicates that the loaded transport end moment is equal to the sum of the loaded transport start moment and the load transport time. Eq 20 indicates that the end moment of the task is equal to the sum of the loaded transport end moment and the unloading time. Eq 21 indicates that the processing start moment is not earlier than the completed moment of the transport task of the workpiece, the processing completed moment of the previous task process in this machine, or the completed moment of the previous process of the workpiece in this task. Eq 22 indicates that at most one RGV is allowed at the same station and time. Eq 23 indicates the initial position of RGVs. Eq 24 indicates that no conflict is allowed. Eq 25 indicates the range of values of the variable.

## 4. Multi-objective improved discrete grey wolf optimizer

To solve DRISP-PERCFR, MOID-GWO is elaborated in this section. The improved process mainly includes the following steps: encoding and decoding, generating initial solutions, social hierarchy, prey search mechanism, and local search mechanism.

### 4.1. Encoding

DRISP-PERCFR contains three resources, including workpieces, machines, and RGVs. For ease of expression, we adopt a three-chain coding strategy based on operation-machine-RGV, denoted as *OS*, *MS*, and *TS*, respectively, an example of which is shown in Fig 4.

A virtual process is added after the last process of all the workpieces to express the RGV task of transporting the workpieces to the unloading platform. Thus, the length of each chromosome sequence is $Lg = n + \sum_{i=1}^{n} p_i$.

The value in *OS* represents $N_i$, and the number of occurrences $j$ represents the $j_{th}$ process of $N_i$.

The value of *MS* represents the number of machines assigned to the process, arranged according to the process number of the workpiece. The machine assigned to the virtual process is the equivalent of the machine in the unloading platform.

The value of *TS* represents the RGV assigned when transported from the previous location to the current production facility according to *OS*.

## 4.2. Decoding

Since the encoding does not consider CFRP, a construction algorithm is used to decode the chromosome into a feasible solution.

Firstly, the OS is converted to the corresponding process sequence. The OS is read sequentially from left to right to determine $O_{ij}$, $M_k$, $t_{ijk}$, $C_{i(j-1)k}$ for the previous process $O_{i(j-1)}$, $R_v$, $STT_{ijv}$.

Secondly, the sequence of tasks performed by each RGV is derived from the chromosomes. Taking the encoding in Fig 4 as an example, the task sequence $rs_1 = \{O_{11}, O_{21}, O_{31}, O_{12}, O_{32}, O_{33}\}$, $rs_2 = \{O_{22}, O_{13}, O_{23}, O_{14}, O_{24}, O_{34}\}$. Next, $rs_1$ and $rs_2$ are transformed by combining the loading and unloading station numbers where the selected machine of each process is located. Therefore, the converted *RS* named $ts_1$ and $ts_2$ is shown in Fig 5.

Finally, CFRP is considered between RGVs. A conflict occurs when Eq 24 is not satisfied. Three methods are used to avoid defined conflicts.

1, Eqs 1 and 2 is used to determine whether the RGV is a loading/unloading task, and if so, the conflict is $EV_1$. At this point, the RGV that are not in the transport task wait to avoid conflicts

$$B_v^{i(i+1)} = V_{v'}^{i'} - (3 - 2v)\theta/l_r \qquad \text{Eq26}$$

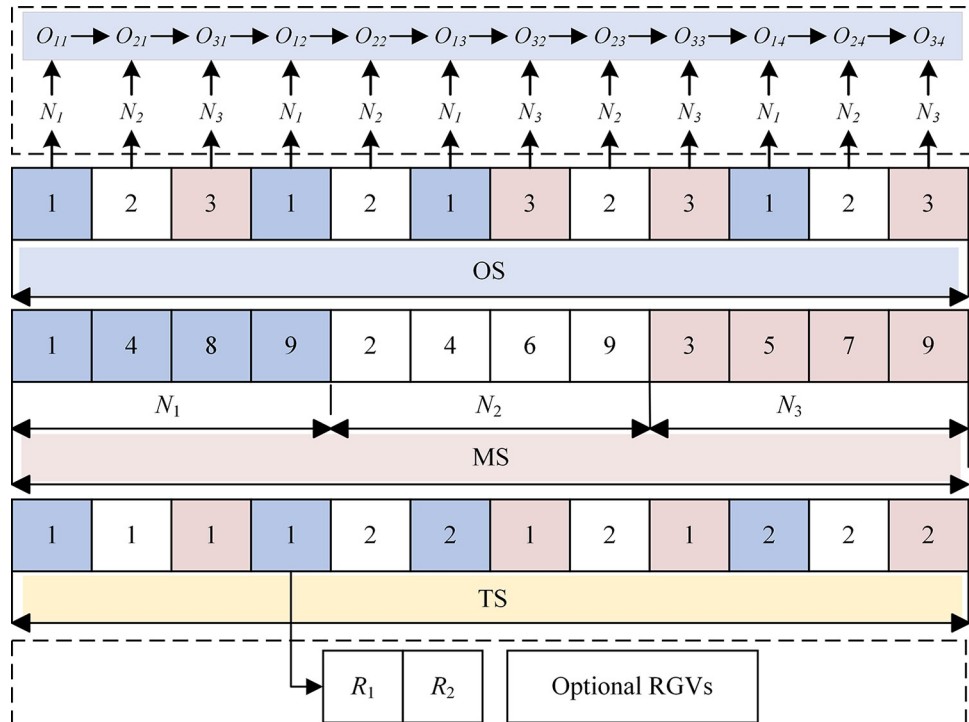

**Fig 4. A three-chain coding strategy based on operation-machine-RGV.**

2, Eq 3 is used to determine whether this RGV is a waiting task, and if so, it is a conflict $EV_2$. At this point, the RGV in the waiting task backs off to avoid conflict.

$$B_v^{i'(i'+1)} = V_v^{i+1} - (3 - 2v)\theta/l_r \qquad\qquad Eq27$$

3, In case of conflicting $EV_3$, the RGV that arrives at the station earlier proceeds as planned and the later RGV waits to avoid conflict.

$$tC'V_v^i = tS_{v'}^{i'} \qquad\qquad Eq28$$

where $tC'V_v^i$ represents the updated departure moment.

The pseudo-code to decode RS by the above three methods is as follows.

```
Algorithm 1: RS decoding considering CFRP
Input: Sequence ts_v of RGV without considering CFRP
Output: Sequence ts'_v of RGV tasks after considering CFRP in decoding
```

1. Calculations $\sum_{v=1}^{2} EV_{ev}$ based on *Def. 1-Def. 3.*

2. While $\sum_{v=1}^{2} EV_{ev} \geq 1$:

```
3.  Switch e:       // Determine the type of conflict
4.    Case 1: Conflict avoidance using Eq 26
5.    Case 2: Conflict avoidance using Eq 27
6.    Case 3: Conflict avoidance using Eq 28
7.    Update ts_v based on the resulting B_v^{ij}8.    Update data on V_v^i in ts_v,
including tSV_v^i tUV_v^i and tCV_v^i9.      ts'_v = ts_v
10. End while
11. return ts'_v
```

## 4.3. Population initialization

To ensure the quality and diversity of the initial population, random initialization, RPTL, and IAPH are used to initialize the population. The sequence of *OS* uses random initialization. Therefore, the MS and TS generation methods are highlighted.

**4.3.1. Sequence initialization of MS based on RPTL.** The RPTL strategy uses a multi-stage decision-making approach to improve initial population quality.

Firstly, a disjunctive diagram is created for transport time and loading/unloading time as shown in Fig 6. The problem consists of 3 workpieces and a total of 9 processes. There are associated weights on the arc. The first weight represents the loading/unloading time of the process at the previous machine. The second weight represents the transport time between the previous and current machines. As an example, there are 3 available machines for the $O_{21}$. Thus $[3\ 3\ 6]^T$ represents the transport time from the loading platform to the $O_{21}$ optional machines. $[6\ 3\ 3]$ indicates the transport time of the optional machine from $O_{23}$ to the unloading platform. In addition, $[6\ 6\ 13]^T$ denotes the loading/unloading time of $O_{21}$.

Secondly, based on Fig 6, a directed acyclic diagram of the transport process is created, as shown in Fig 7. The vertices represent optional devices for the process, where $M_4$ and $M_5$ are two optional machines for $O_{22}$. The weights on the arcs indicate the transport time between machines. The circled number indicates the loading/unloading time required for the process. There are 18 paths in Fig 7 and the path with the shortest time is the best machine selection option for $N_2$.

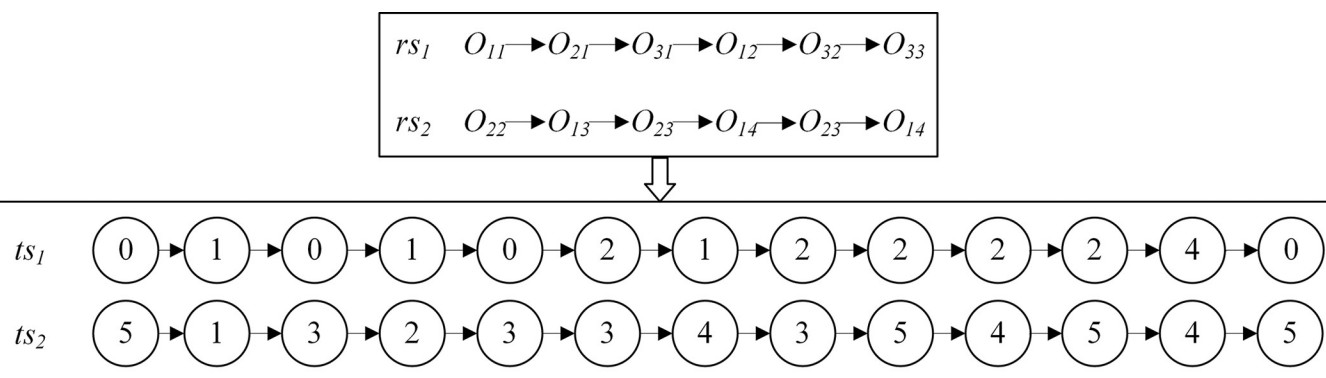

**Fig 5. Converting RGV task sequences to related routes.**

Finally, the processing time is assigned to vertices, and the directed acyclic diagram is divided into 5 stages as shown in Fig 8. The value of the vertex, marked in bold, is the sum of the previous vertex and the arc. RPTL selects the machine with the smallest value at each stage. Path 1 is $M_0$-$M_1$-$M_4$-$M_6$-$M_9$, which takes 175min. path 2 is $M_0$-$M_1$-$M_5$-$M_6$-$M_9$, which also takes 175min. Although the path selected by RPTL is not necessarily the true optimal path, RPTL can simultaneously reduce the transport time and loading/unloading time of the work-pieces, and it can improve the diversity of the initial solutions.

**4.3.2. Sequence initialization of TS based on IAPH.** The full details of APH are described in [31]. According to the characteristics of *TS*, the sequence initialization of *TS* based on IAPH is as follows:

Step 1: Each RGV sends a transport proposal request to the unassigned transport tasks and selects the transport task with the smallest transport time as a candidate task.

Step 2: Calculate the RGV transport time and send the transport task request.

Step 3: The transport task receiving the proposal sends an acknowledgment message to the RGV with the shortest distance.

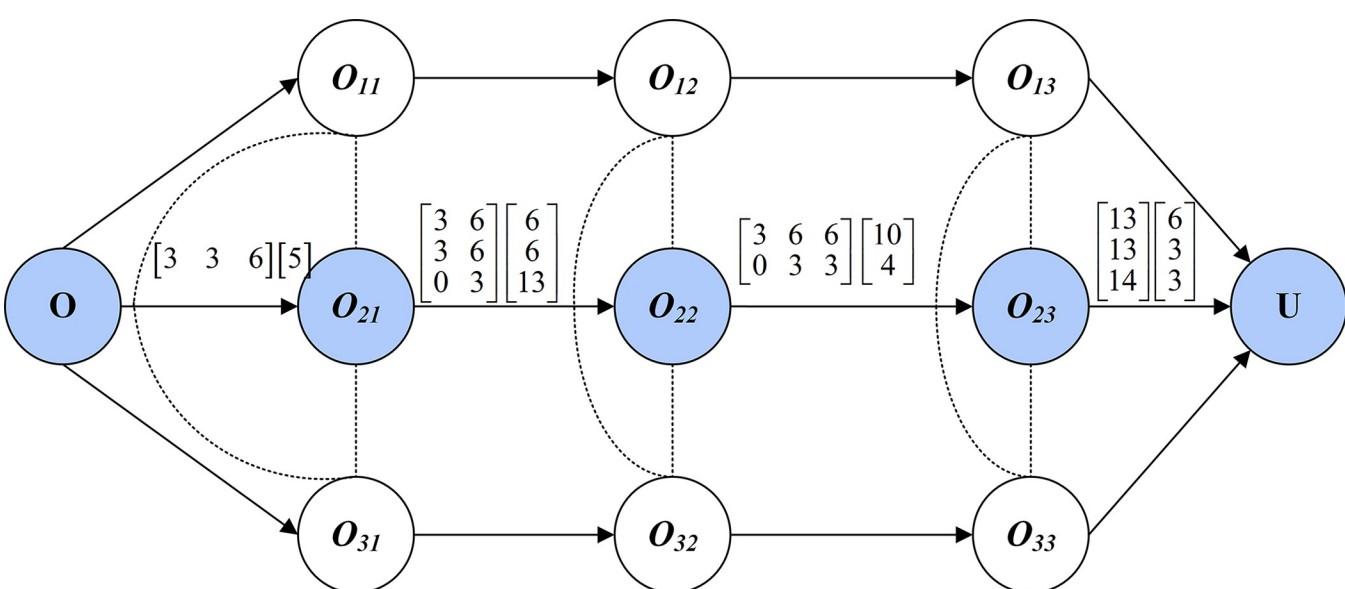

**Fig 6. Disjunctive diagram of transport time and loading/unloading time.**

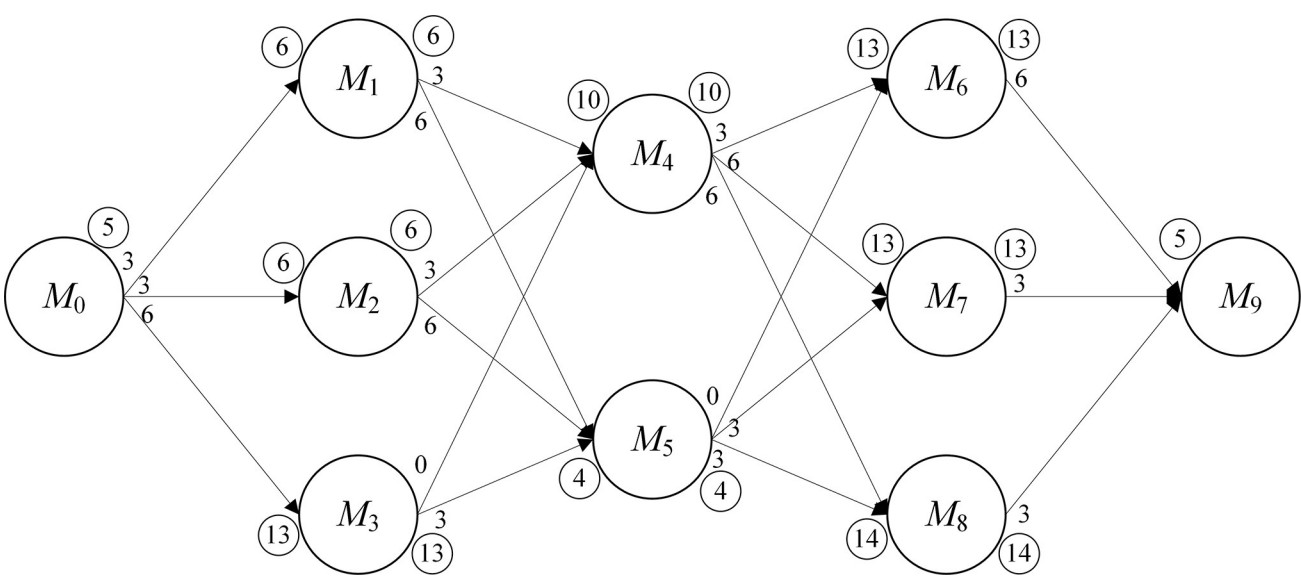

**Fig 7. Directed acyclic diagram of the transport process.**

Repeat the above steps until all transport tasks are assigned to the appropriate RGV.

## 4.4. Social hierarchy

The solution of a multi-objective optimization problem is usually a set of non-dominated solutions. However, GWO is mostly used to solve single-objective optimization problems, so the population is ranked according to the size of crowding and the optimal three individuals are selected. Crowding $D_g$ for the individual is calculated as follows:

$$D_g = \{ \begin{array}{l} \infty, if g = 1 \ or \ g = l \\ |f_1(g+1) - f_1(g-1)| + |f_2(g+1) - f_2(g-1)| + |f_3(g+1) - f_3(g-1)|, \ otherwise \end{array} \qquad \text{Eq29}$$

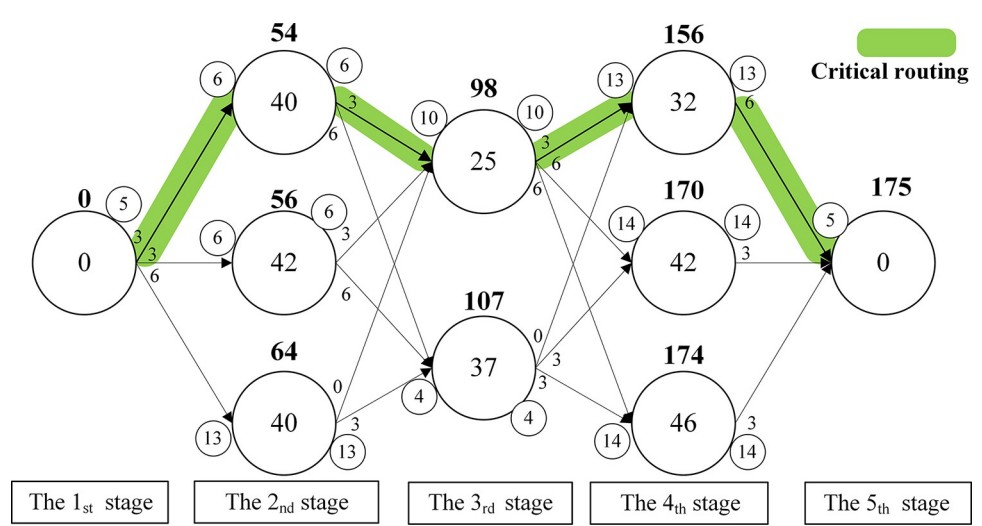

**Fig 8. Directed acyclic diagram.**

where $g$+1 and $g$-1 are individuals adjacent to $g$ in the objective space. $l$ is the number of individuals. $f_1(g)$, $f_2(g)$, and $f_3(g)$ represent the three objective function values of $g$, respectively.

The first three historically optimal solutions $\alpha$, $\beta$, and $\delta$ are obtained by the following assumptions.

1. If there are more than 2 solutions in the first frontier of the archive, choose 3 individuals at random and compute the crowding of the three individuals, with the largest being $\alpha$, the second $\beta$, and the rest $\delta$.

2. If there are only 2 solutions in the first frontier, randomly choose $\alpha$ and $\beta$, and randomly select $\delta$ in the second frontier.

3. If there is only 1 solution in the first frontier, that solution is $\alpha$. If there are more than 2 solutions in the second frontier, from which $\beta$ and $\delta$ are chosen at random.

4. If there is only 1 solution in the first frontier, the solution is $\alpha$. If there is also only 1 solution in the second frontier, the solution is $\beta$. Randomly select $\delta$ in the third frontier.

## 4.5. Prey search mechanism

The problem is a discrete optimization problem, but GWO is mainly used to solve continuous optimization problems. To ensure the feasibility of the solution, a new search operator is proposed to update the location of individual grey wolves in MOID-GWO. The position update equation is as follows:

$$X_{new}(t) = \begin{cases} Cro(X(t), X_\alpha(t)), & rand < \dfrac{1}{3} \\ Cro(X(t), X_\beta(t)), & \dfrac{1}{3} < rand < \dfrac{2}{3} \\ Cro(X(t), X_\delta(t)), & otherwise \end{cases} \qquad \text{Eq30}$$

where $rand$ is a random number between 0 and 1 obeying a uniform distribution, the function $Cro()$ represents the crossover operation.

Since each grey wolf has three layers of chromosomes, different search mechanisms are designed. Precedence operation crossover (POX) [32] is used for $OS$ and improved multi-point precedence operation crossover (IMPX) for $MS$ and $TS$. The crossover will result in two children individuals. The better of the two individuals is chosen as the new one based on the dominant relationship. If two individuals do not dominate each other, one of them is chosen as the new individual.

The following describes the POX-based $OS$ search mechanism, and the IMPX-based $MS$ and $TS$ search mechanisms, respectively.

**4.5.1. POX-based OS search mechanism.** *Parent*1 and *Parent*2 represent the two paternal chromosomes, and *Children*1 and *Children*2 are the two child chromosomes resulting from crossover. The POX-based $OS$ search mechanism is shown in Fig 9. Randomly divide all the workpieces into two non-empty workpiece sets $N^1$ and $N^2$, where $N^1 \cup N^2 = N$. Copy workpieces contained in $N^1$ by *Parent*1 to *Children*1, and workpieces contained in $N^1$ by *Parent*2 to *Children*2, retaining the original positions. Copy workpieces contained in $N^2$ by *Parent*2 to *Children*1, and copy workpieces contained in $N^2$ by *Parent*1 to *Children*2, preserving the original sequence.

**4.5.2 IMPX-based MS and TS search mechanisms.** First, a vector $R = [r_1, r_2, \ldots, r_{Lg}]$ is randomly generated, where $r_i$ is a random number that obeys a uniform distribution between 0 and 1. If $r_i \leq pf$, record the index of $r_i$. Select the position corresponding to $r_i$ in *Parent1* and

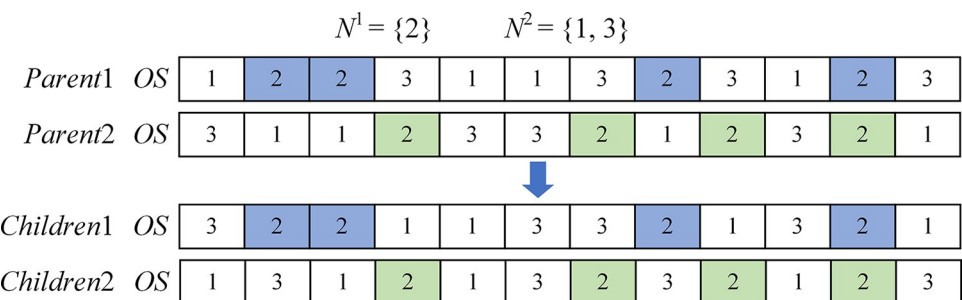

**Fig 9. POX-based OS search mechanism.**

*Parent2*, exchange the assigned production devices, and keep the other devices to the children to generate the children *Children1* and *Children2*. *pf* is calculated as follows:

$$pf = pf_{\max} - \frac{pf_{\max} - pf_{\min}}{iter} \times iterMax \qquad \text{Eq31}$$

An example of IMPX-based *MS* and *TS* search mechanisms is shown in Fig 10(A) and 10 (B), respectively, where *pf* = 0.4.

## 4.6. Local search mechanism

Two neighbor search operators in MOID-GWO are used to achieve local search, an example is shown in Fig 11.

*N*(1): Two different genes are randomly selected on *OS*, and the first is moved one bit after the second gene locus, as shown in Fig 11(A).

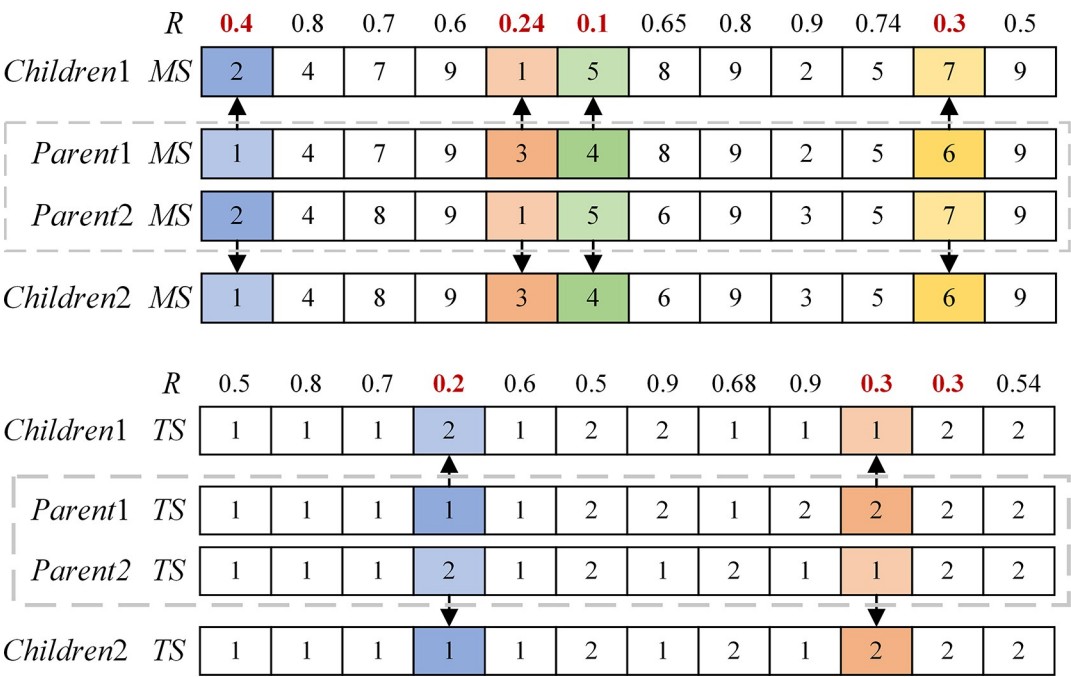

**Fig 10.** An example of IMPX-based search mechanisms:(a) MS, (b) TS.

*N*(2): Three different genes are randomly selected on *OS* and one optimal neighbor solution is taken as the new solution in all their neighbors, as shown in Fig 11(B).

Since the local search mechanism takes a long time, MOID-GWO performs a local search for all non-inferior solutions of the first frontier every 20 iterations.

### 4.7. Algorithm process

As mentioned above, the basic flowchart of MOID-GWO is shown in Fig 12.

The steps are as follows:

Step 1: Set the algorithm parameters.

Step 2: Encoding and decoding, as detailed in Section 4.1 and Section 4.2.

Step 3: Initialize the population, as detailed in Section 4.3.

Step 4: Rank of populations, as detailed in Section 4.4.

Step 5: Check whether the maximum number of iterations is satisfied, if yes, perform Step 11; otherwise, perform Step 6.

Step 6: Select $\alpha$, $\beta$ and $\delta$, see Section 4.4 for details.

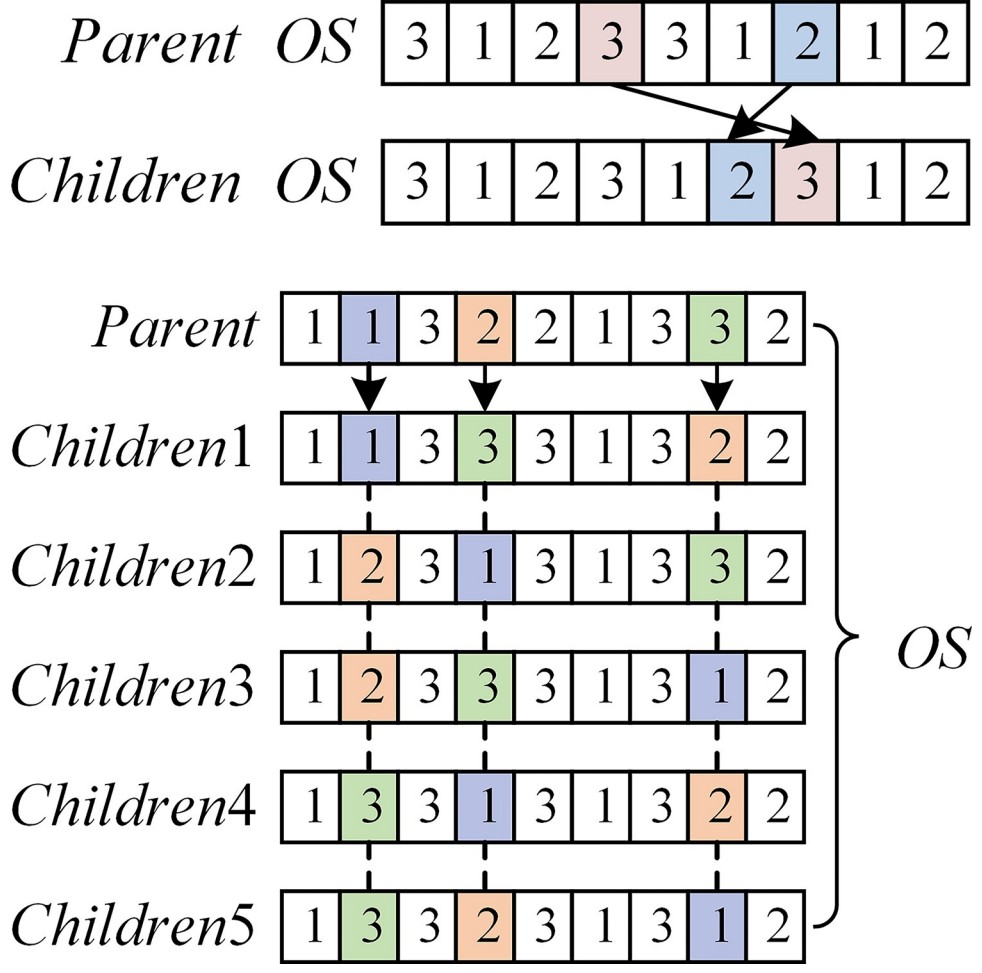

**Fig 11.** Examples of neighbor search operators: (a) N(1), (b) N(2).

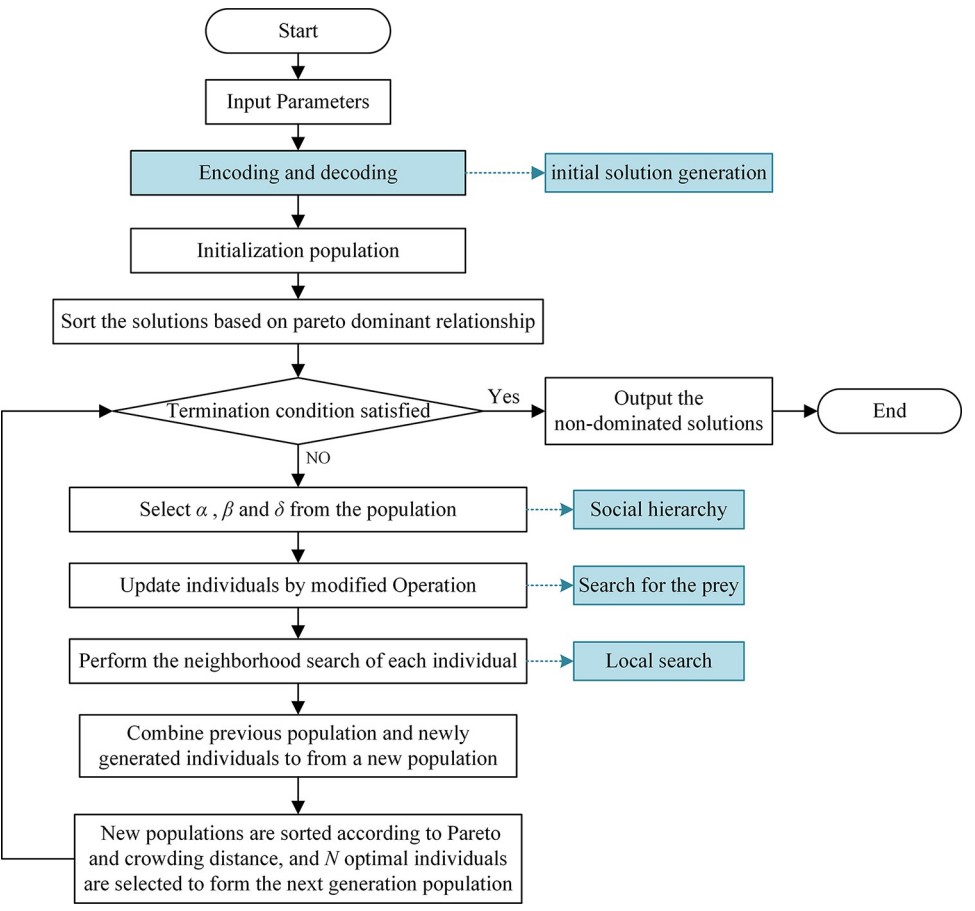

**Fig 12. The basic flowchart of MOID-GWO.**

Step 7: Update the location of individuals based on the improved prey search mechanism, as detailed in Section 4.5.

Step 8: Update the locations of individuals using the designed local search strategy to generate a new population, as detailed in Section 4.6.

Step 9: Merge the new and old populations into a new population.

Step 10: Sort the new population according to Pareto and congestion, select N optimal individuals to form the next generation population, and return to perform Step 5.

Step 11: Output non-dominated solutions.

## 5. Numerical results

In this section, the performance of the proposed MOID-GWO algorithm is evaluated by constructing test instances and comparing them with other optimization algorithms.

### 5.1. Test instances

According to the above design of different scale examples to experiments. The distribution of randomly generated data is shown in Tables 2 and 3. Details can be found in support information S1 and S2 Tables. Each process is performed by parallel machines [33].

**Table 2. Data set distribution.**

| Input variable | Types | Distribution | Comment |
|---|---|---|---|
| $n$ | - | 5,10,20,40,60,80 | - |
| $m$ | - | 5,6,7,8,9,10,11 | - |
| Process1 | Production time (min) | U(30,50) | Parallel machine |
| | Loading and unloading time (min) | U(6,14) | |
| Process2 | Production time (min) | U(20,40) | Parallel machine |
| | Loading and unloading time (min) | U(2,10) | |
| Process3 | Production time (min) | U(30,60) | Parallel machine |
| | Loading and unloading time (min) | U(5,15) | |

The set of optional machines is represented as the machine corresponding to the process.

For ease of illustration, we number the examples in $n\#m\#$ format. $n10m8$ means 10 workpieces and 8 machines. In addition, for all instances, $v_R$ = 5m/min, $\theta$ = 15m, the loading and unloading times of the workpieces at the loading and unloading platform are 5min, and the distance between the loading and unloading points of the two machines is 15m.

## 5.2. Effectiveness analysis of the improvement strategy

To evaluate the effectiveness of the proposed RPTL strategy, we conducted a comparison experiment between MOID-GWO and MOID-GWO$_r$ (MOID-GWO using a random initial population strategy). In MOID-GWO$_r$, other parameters are consistent except the initial population strategy.

Since spread(SP), generational distance(GD) and inverted GD(IGD) can effectively evaluate the diversity and overall performance of multi-objective evolutionary algorithms (MOEAs), these three metrics are used to evaluate the effectiveness of RPTL strategies.

Two algorithms are independently run 30 times on 42 instances of different scales, and the results of average value are shown in Table 4. The best values of each metric in different algorithms are shown in bold.

It can be seen from the results that MOID-GWO obtained the best value of SP in 29 instances, indicating that the strategy can effectively maintain the diversity of the population. MOID-GWO obtains the best value of GD in 32 instances, which means that MOID-GWO has better convergence than MOID-GWO$_r$ in most instances. Most notably, MOID-GWO obtains the best value of IGD in 36 instances, which shows that MOID-GWO is superior to MOID-GWOr in diversity and convergence.

**Table 3. Set of optional machine.**

| $m$ | Process | | |
|---|---|---|---|
| | Process1 | Process 2 | Process 3 |
| 5 | [$M_1$-$M_2$] | [$M_3$] | [$M_4$-$M_5$] |
| 6 | [$M_1$-$M_2$] | [$M_3$-$M_4$] | [$M_5$-$M_6$] |
| 7 | [$M_1$-$M_2$] | [$M_3$-$M_4$] | [$M_5$-$M_7$] |
| 8 | [$M_1$-$M_3$] | [$M_4$-$M_5$] | [$M_6$-$M_8$] |
| 9 | [$M_1$-$M_3$] | [$M_4$-$M_5$] | [$M_6$-$M_9$] |
| 10 | [$M_1$-$M_3$] | [$M_4$-$M_5$] | [$M_6$-$M_9$] |
| 11 | [$M_1$-$M_4$] | [$M_5$-$M_7$] | [$M_8$-$M_{11}$] |

**Table 4. Effectiveness analysis of RPTL.**

| Problem | SP | | GD | | IGD | |
|---|---|---|---|---|---|---|
| | MOID-GWO$_r$ | MOID-GWO | MOID-GWO$_r$ | MOID-GWO | MOID-GWO$_r$ | MOID-GWO |
| n5m5 | 1.18E-01 | **1.05E-01** | 3.14E-02 | **1.08E-02** | 4.52E-02 | **4.23E-02** |
| n10m5 | 1.23E+00 | **8.96E-01** | 7.41E-02 | **7.02E-02** | 1.43E-02 | **1.02E-02** |
| n20m5 | **6.02E-01** | 6.72E-01 | 6.87E-02 | **6.49E-02** | **1.33E-02** | 4.78E-02 |
| n40m5 | 1.37E-01 | **1.22E-01** | **2.29E-02** | 2.45E-02 | 4.81E-02 | **1.95E-02** |
| n60m5 | **8.73E-01** | 9.39E-01 | **7.80E-02** | 8.90E-02 | 4.94E-02 | **3.49E-02** |
| n80m5 | **9.61E-01** | 1.16E+00 | 8.93E-02 | **6.29E-02** | 1.45E-02 | **2.46E-02** |
| n5m6 | 8.91E-01 | **8.56E-01** | 3.22E-02 | **1.15E-02** | 7.89E-02 | **3.62E-02** |
| n10m6 | **6.65E-01** | 7.32E-01 | 2.84E-02 | **2.23E-02** | 1.46E-01 | **2.63E-02** |
| n20m6 | 1.39E+00 | **8.29E-01** | **2.25E-02** | 4.36E-02 | **6.00E-02** | 7.36E-02 |
| n40m6 | 7.55E-01 | **6.87E-01** | 5.76E-02 | **5.08E-02** | 5.67E-02 | **4.83E-02** |
| n60m6 | 8.87E-01 | **8.59E-01** | 3.90E-02 | **2.31E-02** | 8.15E-02 | **7.68E-02** |
| n80m6 | **6.01E-01** | 7.05E-01 | **4.35E-02** | 5.11E-02 | 1.75E-02 | **1.21E-02** |
| n5m7 | 8.92E-01 | **8.21E-01** | 6.30E-02 | **4.55E-02** | 5.40E-02 | **3.57E-02** |
| n10m7 | 3.34E-01 | **2.65E-01** | **3.86E-02** | 4.05E-02 | **7.81E-03** | 8.86E-03 |
| n20m7 | **6.02E-01** | 6.12E-01 | 3.47E-02 | **2.58E-02** | 4.43E-02 | **3.81E-02** |
| n40m7 | 6.71E-01 | **5.79E-01** | **8.29E-02** | 8.72E-02 | 9.53E-02 | **2.85E-03** |
| n60m7 | 9.20E-01 | **7.42E-01** | **5.28E-02** | 6.12E-02 | 7.24E-02 | **7.08E-02** |
| n80m7 | 4.84E-01 | **4.52E-01** | 2.95E-02 | **2.83E-02** | 9.64E-02 | **2.49E-03** |
| n5m8 | **8.44E-01** | 8.16E-01 | **2.66E-02** | 2.16E-03 | 8.92E-02 | **8.44E-02** |
| n10m8 | 0.84E-01 | **7.90E-01** | 5.89E-02 | **5.16E-02** | 8.47E-02 | **5.60E-02** |
| n20m8 | 6.45E-01 | **5.78E-01** | 4.99E-02 | **2.52E-02** | 6.81E-02 | **6.21E-01** |
| n40m8 | **6.12E-01** | 6.43E-01 | 2.48E-02 | **2.40E-02** | **5.02E-03** | 5.59E-02 |
| n60m8 | 8.83E-01 | **7.80E-01** | 5.99E-02 | **5.46E-02** | 7.59E-02 | **8.60E-02** |
| n80m8 | 7.85E-01 | **6.21E-01** | 7.54E-02 | **7.05E-02** | 9.04E-02 | **8.11E-02** |
| n5m9 | **8.05E-01** | 8.20E-01 | **6.38E-02** | 9.97E-02 | 1.40E-02 | **2.55E-02** |
| n10m9 | 8.01E-01 | **7.18E-01** | **3.49E-02** | 4.49E-02 | 1.04E-02 | **1.11E-02** |
| n20m9 | 8.06E-01 | **7.30E-01** | 8.26E-02 | **7.72E-02** | 5.32E-03 | **4.15E-03** |
| n40m9 | 1.14E+00 | **8.28E-01** | **5.75E-02** | 6.72E-02 | 6.36E-02 | **8.07E-02** |
| n60m9 | **7.02E-01** | 9.21E-01 | **1.04E-02** | 2.51E-02 | **2.00E-02** | 3.21E-02 |
| n80m9 | 5.57E-01 | **5.09E-01** | 9.61E-03 | **8.84E-03** | 2.01E-02 | **1.50E-02** |
| n5m10 | 5.85E-01 | **4.55E-01** | 8.92E-03 | **7.74E-03** | 2.41E-02 | **1.07E-02** |
| n10m10 | **8.28E-01** | 9.28E-01 | 6.72E-02 | **5.63E-02** | 4.63E-02 | **3.73E-02** |
| n20m10 | **4.51E-01** | 9.10E-01 | 5.94E-02 | **4.53E-02** | **6.47E-02** | 8.76E-02 |
| n40m10 | 7.58E-01 | **5.48E-01** | 6.19E-02 | **5.09E-02** | 9.42E-02 | **7.12E-02** |
| n60m10 | 9.24E-01 | **7.17E-01** | **4.28E-02** | 5.71E-02 | 3.99E-02 | **3.78E-03** |
| n80m10 | 9.10E-01 | **7.75E-01** | 4.22E-02 | **2.78E-02** | 6.72E-02 | **5.41E-02** |
| n5m11 | 1.04E+00 | **8.38E-01** | 9.16E-02 | **8.38E-02** | 4.37E-02 | **4.23E-02** |
| n10m11 | 8.31E-01 | **7.60E-01** | 3.88E-02 | **3.53E-02** | 1.49E-02 | **1.46E-02** |
| n20m11 | 8.92E-01 | **7.08E-01** | 6.19E-02 | **5.14E-02** | 1.02E-02 | **7.59E-03** |
| n40m11 | **8.33E-01** | 9.55E-01 | 3.45E-02 | **3.23E-02** | 3.10E-03 | **2.50E-03** |
| n60m11 | 8.45E-01 | **6.92E-01** | 7.31E-01 | **6.98E-02** | 1.72E-02 | **1.33E-02** |
| n80m11 | 9.22E-01 | **7.21E-01** | 2.15E-02 | **1.50E-02** | 5.22E-02 | **3.38E-02** |
| Hit rate | 13/42 | 29/42 | 10/42 | 32/42 | 6/42 | 36/42 |

In sum, RPTL initializing population strategy is helpful to improve the accuracy of the algorithm Effectiveness analysis showed that RPTL can help the MOID-GWO improve the quality of the initial solution, and achieve better accuracy ultimately.

## 5.3. Experimental comparisons

To evaluate the effectiveness of the proposed MOID-GWO, MOEAs, including NSGA-II [34], SPEA2 [35], and MOPSO [36], are used for comparison.

All MOEAs use the same coding and decoding methods. The same crossover operator as MOID-GWO is used in NSGA-II and SPEA2. Other parameter settings used for comparison are the same as in the original literature.

In addition to the SP, GD and IGD mentioned above, the metric of convergence and diversity (CD) is used, which can visualize the convergence curve [37].

30 independent tests are performed on each instance separately and the statistical results obtained are shown in Table 5.

The rows at the bottom of the table list the key performance ratios for the corresponding indicators. For example, 3/42 means that the corresponding algorithm outperforms the other algorithms in 3 out of 42 problems for the given metrics.

MOID-GWO outperforms the other comparison algorithms in most cases. Especially in GD and IGD, the advantage of MOID-GWO is obvious. In SP, MOID-GWO is better than other MOEAs.

Due to the stochastic nature of the model, statistical tests are performed. Wilcoxon signed-rank test [38] is used to detect significant differences between the results of MOEAs. Confidence level for all experiments is 95% ($\alpha = 0.05$). The $p$-values for the Wilcoxon signed rank test are shown in Table 6. The results showed that MOID-GWO is significantly better than NSGA-II, SPEA2, and MPOSPO in terms of GD and IGD.

To visualize the results, randomly selected small, medium, and large-scale instances ($n10m8$, $n40m8$, and $n80m8$). Fig 13 shows the boxplot of IGD of MOEAs in 30 runs, which proves the superiority of MOID-GWO. Fig 13(A)–13(C) represent the results of n10m8, n40m8 and n80m8 respectively.

Further, a CD metric is used to analyze the convergence of the algorithm. The convergence curves obtained by MOEAs are shown in Fig 14. Fig 14(A)–14(C) represent the results of n10m8, n40m8 and n80m8 respectively. It can be seen that for small and medium-scale instances, MOID-GWO can jump out of the local optimum at the later stage of iteration. For a large instance, although MOID-GWO converges slower than other algorithms, it can find better non-dominated solutions.

The results with the best IGD in MOEAs are selected and visualized, as shown in Fig 15. Fig 15(A)–15(C) represent the results of n10m8, n40m8 and n80m8 respectively. It can be seen that the Pareto frontier obtained by MOID-GWO are all close to the Pareto optimum compared to other algorithms, proving that MOID-GWO can obtain better non-dominated solutions.

## 5.4. Discussion

The main reasons why MOID-GWO outperforms the other three MOEAs in terms of solution distribution, convergence, and accuracy can be described as follows:

Firstly, the initial population is generated using the RPTL and the random strategy, RPTL improves the quality of the initial population and facilitates the search for the optimal solution. The random strategy keeps good diversity in the population.

**Table 5. The result of metrics including SP, GD and IGD obtained by MOEAs.**

| Problem | MOID-GWO | | | NSGA-II | | | SPEA2 | | | MOPSO | | |
|---|---|---|---|---|---|---|---|---|---|---|---|---|
| | SP | GD | IGD | SP | GD | IGD | SP | GD | IGD | SP | GD | IGD |
| $n5m5$ | **1.16E+00** | **2.22E-02** | **1.25E-02** | 1.45E+00 | 3.41E-02 | 2.61E-02 | 1.32E+00 | 3.15E-02 | 2.56E-02 | 1.18E+00 | 3.48E-02 | 2.71E-02 |
| $n10m5$ | 7.31E-01 | **2.29E-02** | 2.04E-02 | 7.41E-01 | 3.51E-02 | **1.87E-02** | **6.38E-01** | 4.10E-02 | 2.11E-02 | 8.03E-01 | 4.40E-02 | 2.80E-02 |
| $n20m5$ | 8.60E-01 | **1.57E-02** | 2.84E-02 | 8.73E-01 | 3.16E-02 | **2.25E-02** | **7.87E-01** | 2.96E-02 | 3.26E-02 | 8.50E-01 | 3.17E-02 | 3.19E-02 |
| $n40m5$ | **8.45E-01** | **2.05E-02** | **3.77E-02** | 9.07E-01 | 3.03E-02 | 5.23E-02 | 9.49E-01 | 3.22E-02 | 5.04E-02 | 9.26E-01 | 3.87E-02 | 5.51E-02 |
| $n60m5$ | 5.58E-01 | **7.63E-03** | **2.74E-02** | 4.96E-01 | 8.92E-03 | 3.64E-02 | **4.41E-01** | 9.01E-03 | 3.87E-02 | 5.15E-01 | 9.58E-03 | 4.15E-02 |
| $n80m5$ | 7.06E-01 | 2.66E-02 | **2.91E-02** | **6.08E-01** | 2.36E-02 | 3.98E-02 | 7.69E-01 | **1.74E-01** | 3.44E-02 | 7.58E-01 | 2.93E-03 | 3.42E-02 |
| $n5m6$ | **8.11E-01** | **8.72E-03** | **8.49E-03** | 9.34E-01 | 1.08E-02 | 1.11E-02 | 8.90E-01 | 9.61E-03 | 1.43E-02 | 8.32E-01 | 9.64E-03 | 9.67E-03 |
| $n10m6$ | **8.68E-01** | **2.11E-02** | **7.49E-03** | 1.21E+00 | 3.49E-02 | 1.12E-02 | 9.75E-01 | 4.51E-02 | 9.17E-03 | 9.18E-01 | 3.57E-02 | 1.02E-02 |
| $n20m6$ | 8.03E-01 | **1.29E-02** | 6.76E-02 | 8.39E-01 | 3.49E-01 | 6.02E-02 | 9.28E-01 | 2.73E-02 | **5.57E-02** | 9.27E-01 | 3.26E-02 | 6.89E-02 |
| $n40m6$ | 6.03E-01 | **1.04E-02** | 2.07E-02 | 7.15E-01 | 1.99E-02 | 3.51E-02 | 6.08E-01 | 2.17E-02 | 3.63E-02 | 7.27E-01 | 2.21E-02 | 4.18E-02 |
| $n60m6$ | **9.37E-01** | **4.39E-02** | **3.04E-02** | 8.33E-01 | 5.13E-02 | 4.91E-02 | 1.13E+00 | 5.07E-01 | 5.29E-02 | 1.07E+00 | 5.34E-02 | 4.51E-02 |
| $n80m6$ | **7.48E-01** | **2.29E-02** | **9.52E-03** | 7.21E-01 | 3.13E-01 | 1.52E-02 | 8.97E-01 | 3.10E-01 | 1.44E-02 | 8.24E-01 | 3.40E-02 | 1.28E-02 |
| $n5m7$ | **6.30E-01** | 3.57E-02 | 2.56E-02 | 7.90E-01 | 4.16E-02 | 1.96E-02 | 7.18E-01 | 5.43E-02 | 2.95E-02 | 6.37E-01 | 4.32E-02 | **1.31E-02** |
| $n10m7$ | 7.64E-01 | **1.87E-02** | **1.85E-02** | 8.30E-01 | 3.03E-02 | 3.90E-02 | 8.15E-01 | 3.22E-02 | 3.79E-02 | 9.37E-01 | 2.98E-02 | 3.14E-02 |
| $n20m7$ | 8.26E-01 | 4.69E-02 | 3.63E-02 | 8.83E-01 | 6.27E-01 | **2.15E-02** | 9.70E-01 | 5.91E-02 | 3.53E-02 | 9.74E-01 | 5.93E-02 | 3.27E-02 |
| $n40m7$ | **6.29E-01** | **1.56E-02** | **1.35E-02** | 7.96E-01 | 1.95E-02 | 3.48E-02 | 8.16E-01 | 2.16E-02 | 3.05E-02 | 7.62E-01 | 2.90E-02 | 3.75E-02 |
| $n60m7$ | **6.80E-01** | **1.49E-02** | **2.79E-02** | 8.09E-01 | 2.56E-02 | 4.33E-02 | 7.36E-01 | 2.48E-02 | 4.03E-02 | 7.82E-01 | 3.17E-02 | 3.97E-02 |
| $n80m7$ | 7.48E-01 | **1.14E-02** | **5.11E-02** | 7.80E-01 | 2.20E-02 | 7.05E-02 | **7.01E-01** | 2.96E-02 | 7.37E-02 | 7.96E-01 | 2.73E-02 | 7.71E-02 |
| $n5m8$ | **6.04E-01** | **4.16E-02** | **2.59E-02** | 7.03E-01 | 5.85E-02 | 3.80E-02 | 7.48E-01 | 5.64E-02 | 3.65E-02 | 7.87E-01 | 5.73E-02 | 3.97E-02 |
| $n10m8$ | 8.02E-01 | **1.15E-02** | **2.79E-02** | 7.52E-01 | 2.63E-02 | 3.82E-02 | 1.08E+00 | 2.33E-02 | 3.94E-02 | 8.31E-01 | 2.82E-02 | 3.58E-02 |
| $n20m8$ | 9.04E-01 | **6.32E-03** | **2.72E-02** | 1.40E+00 | 8.95E-03 | 3.68E-02 | 9.32E-01 | 8.01E-03 | 3.56E-02 | 1.35E+00 | 8.08E-03 | 3.28E-02 |
| $n40m8$ | **8.01E-01** | 3.09E-02 | **3.24E-02** | 1.06E+00 | 4.21E-02 | 4.42E-02 | 1.38E+00 | **2.40E-02** | 4.67E-02 | 9.56E-01 | 3.22E-02 | 5.09E-02 |
| $n60m8$ | **7.09E-01** | **1.38E-02** | **1.72E-02** | 8.63E-01 | 2.90E-02 | 2.46E-02 | 8.55E-01 | 2.11E-02 | 2.57E-02 | 8.60E-01 | 2.58E-02 | 2.37E-02 |
| $n80m8$ | **5.79E-01** | 2.59E-02 | **1.77E-02** | 6.01E-01 | 4.32E-02 | 2.58E-02 | 7.45E-01 | 4.96E-02 | 2.47E-02 | 6.19E-01 | 3.93E-02 | 2.73E-02 |
| $n5m9$ | 9.43E-01 | **1.21E-02** | 5.95E-02 | 9.18E-01 | 2.97E-02 | 7.27E-02 | 9.50E-01 | 2.86E-02 | 7.06E-02 | **9.03E-01** | 2.29E-02 | 7.67E-02 |
| $n10m9$ | **6.33E-01** | **7.85E-03** | **2.31E-02** | 7.82E-01 | 9.73E-03 | 3.04E-02 | 8.22E-01 | 9.22E-03 | 3.11E-02 | 8.14E-01 | 8.84E-03 | 3.74E-02 |
| $n20m9$ | **8.79E-01** | **2.55E-02** | **1.80E-02** | 9.71E-01 | 3.47E-02 | 3.17E-02 | 9.43E-01 | 3.53E-02 | 2.34E-02 | 9.67E-01 | 3.03E-02 | 3.61E-02 |
| $n40m9$ | **8.30E-01** | 3.05E-02 | **3.26E-02** | 9.24E-01 | 3.60E-02 | 5.09E-02 | 9.68E-01 | **2.12E-02** | 5.63E-02 | 9.86E-01 | 2.91E-02 | 5.27E-02 |
| $n60m9$ | 9.51E-01 | **5.05E-03** | **1.09E-02** | **8.04E-01** | 7.95E-03 | 1.96E-02 | 9.77E-01 | 8.09E-03 | 2.74E-02 | 8.94E-01 | 7.56E-03 | 2.38E-02 |
| $n80m9$ | **9.42E-01** | **2.33E-02** | 3.22E-02 | 1.41E+00 | 3.12E-02 | 3.20E-02 | 9.89E-01 | 4.26E-02 | **2.51E-02** | 1.15E+00 | 3.53E-02 | 3.59E-02 |
| $n5m10$ | 8.74E-01 | **1.82E-02** | 1.70E-02 | 8.46E-01 | 2.64E-02 | 1.06E-02 | **7.26E-01** | 2.96E-02 | **9.23E-03** | 8.53E-01 | 2.71E-02 | 1.37E-02 |
| $n10m10$ | 6.54E-01 | **2.05E-02** | **1.14E-02** | 6.91E-01 | 3.46E-02 | 2.78E-02 | **5.67E-01** | 3.67E-02 | 3.36E-02 | 6.55E-01 | 3.10E-02 | 3.34E-02 |
| $n20m10$ | **6.51E-01** | **1.41E-02** | 2.52E-02 | 8.17E-01 | 2.34E-02 | 3.87E-02 | 8.35E-01 | 2.48E-02 | 3.81E-02 | 7.46E-01 | 2.33E-02 | 3.54E-02 |
| $n40m10$ | 6.22E-01 | **6.27E-02** | 2.15E-02 | **7.05E-01** | 8.92E-03 | 1.50E-02 | 6.54E-01 | 7.85E-03 | **1.29E-02** | 6.29E-01 | 7.51E-03 | 1.59E-02 |
| $n60m10$ | 9.74E-01 | 3.22E-02 | **3.74E-02** | **8.74E-01** | 3.18E-02 | 5.83E-02 | 9.91E-01 | **2.82E-02** | 4.82E-02 | 9.85E-01 | 3.57E-02 | 2.86E-02 |
| $n80m10$ | **8.35E-01** | 2.59E-02 | **2.88E-02** | 9.48E-01 | 2.64E-02 | 4.05E-02 | 1.09E+00 | **1.93E-02** | 4.14E-02 | 9.02E-01 | 2.87E-02 | 4.85E-02 |
| $n5m11$ | **9.36E-01** | **3.77E-02** | **1.76E-02** | 9.51E-01 | 4.68E-02 | 3.64E-02 | 1.43E+00 | 4.23E-02 | 3.99E-02 | 1.12E+00 | 4.35E-02 | 3.38E-02 |
| $n10m11$ | 8.32E-01 | **2.16E-02** | **2.43E-02** | 9.15E-01 | 3.29E-02 | 3.83E-02 | **7.44E-01** | 3.66E-02 | 3.62E-02 | 8.67E-01 | 3.47E-02 | 3.41E-02 |
| $n20m11$ | **7.92E-01** | **7.63E-03** | **8.17E-03** | 8.02E-01 | 8.55E-02 | 1.41E-02 | 8.72E-01 | 8.60E-03 | 1.01E-02 | 8.48E-01 | 9.08E-03 | 1.29E-02 |
| Hit rate | 28/42 | 37/42 | 34/42 | 5/42 | 0/42 | 3/42 | 7/42 | 5/42 | 4/42 | 2/42 | 0/42 | 1/42 |

*Notes*: *Bold indicates that the algorithm yields the best solution for the same metrics.*

Secondly, the modified social hierarchy promotes population diversity, allowing MOID-GWO to quickly converge to the optimal solution and preventing falling into local optima.

**Table 6. Wilcoxon signed-rank test for three metrics. (α = 0.05).**

| Algorithm Comparison Group | SP | | GD | | IGD | |
|---|---|---|---|---|---|---|
| | *p*-value | Sig (*p*<0.05) | *p*-value | Sig (*p*<0.05) | *p*-value | Sig (*p*<0.05) |
| MOID-GWO vs NSGA-II | 1.12E-01 | N | 9.78E-04 | Y | 3.23E-03 | Y |
| MOID-GWO vs SPEA2 | 1.69E-02 | N | 5.96E-03 | Y | 9.78E-04 | Y |
| MOID-GWO vs MOPSO | 2.02E-02 | N | 8.33E-03 | Y | 5.70E-04 | Y |

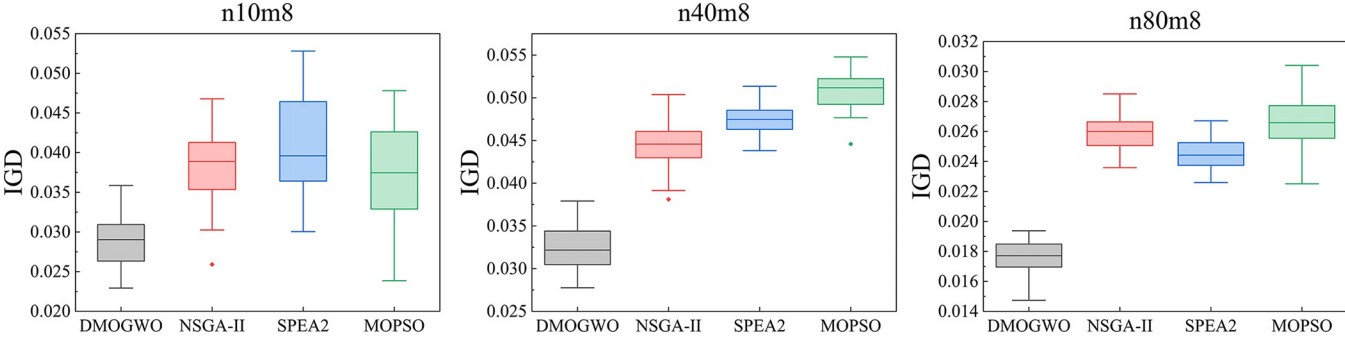

**Fig 13.** Boxplot of IGD obtained by MOEAS: (a) n10m8 problem, (b) n40m8 problem, (c) n80m8 problem.

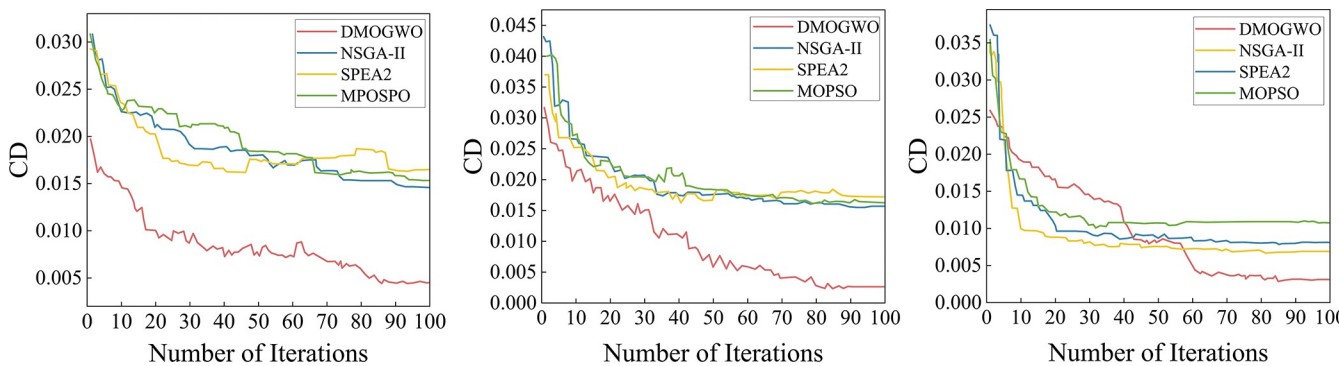

**Fig 14.** CD Convergence curves obtained by MOEAS: (a) n10m8 problem, (b) n40m8 problem, (c) n80m8 problem.

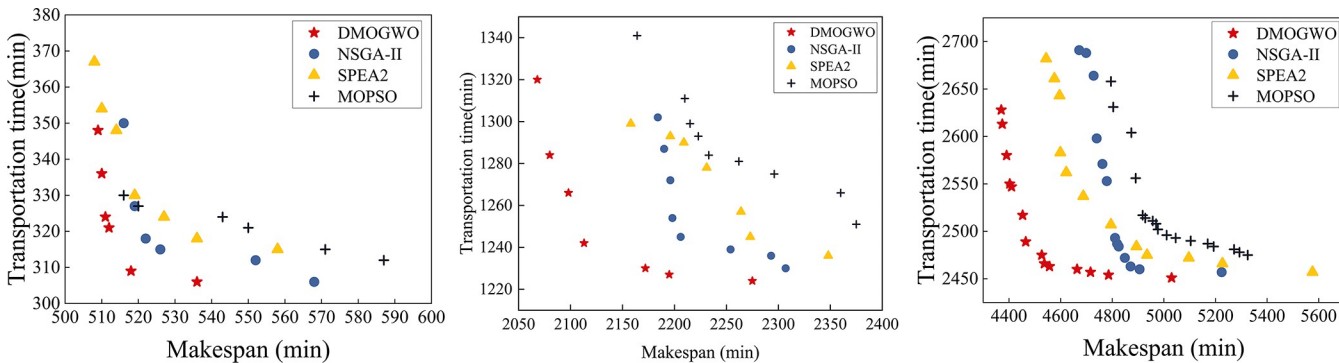

**Fig 15.** Pareto frontier by MOEAs: (a) n10m8 problem, (b) n40m8 problem, (c) n80m8 problem.

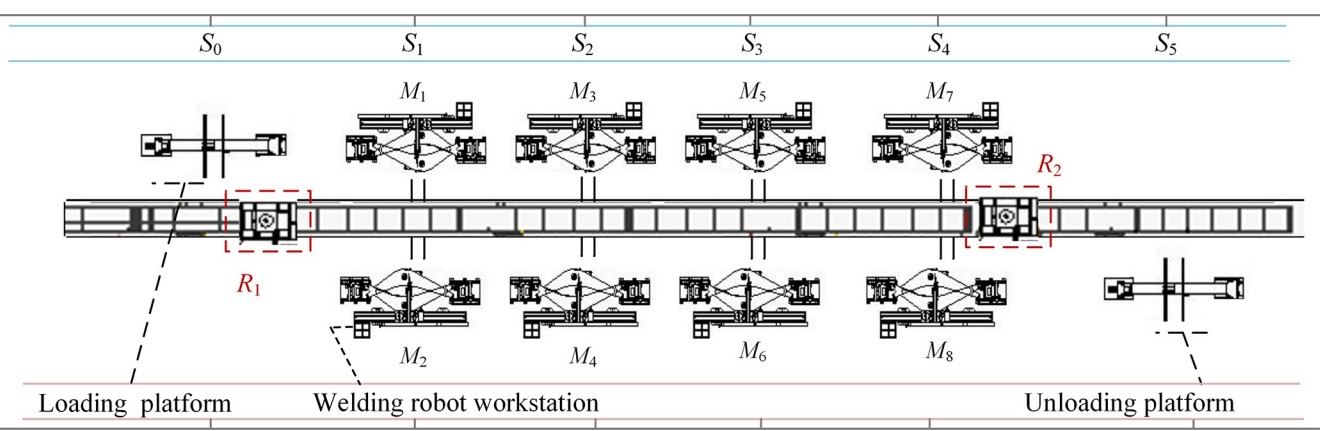

**Fig 16. Welding process site layout.**

Thirdly, the population diversity is maintained using a modified search operator with an embedded crossover operator to enhance the updating mechanism of individual positions.

Finally, the designed neighbor search mechanism extends the search space and gives MOID-GWO better search capability.

## 6. Case study

### 6.1. Background description

In this section, a case study from the production process of welded parts for construction machinery is investigated. Combined with the welding process in the manufacture of an excavator's boom, the welding process is divided into 3 processes, including top and ear plate welding, internal welding, and external welding. Workpieces can be selected for multiple parallel machining in each process. The site layout is shown in Fig 16.

10 types of booms are required to be processed and the data is provided in Tables 7–9, including alternative machines, process times, loading and unloading times, and transport times for each process. Details can be found in support information S3, S4 and S5 Tables respectively.

### 6.2. Problem-solving

The Pareto front obtained by MOID-GWO is shown in Fig 17. Point *A* closest to the origin from the set of Pareto optimal solutions is selected and the corresponding Gantt chart and RGV trajectory are generated.

Since the bi-objective optimization with minimum maximum completion time and minimum transportation time generally does not have a unique optimal solution, it is usually solved for its Pareto optimal solution set, to obtain a specific number to facilitate intuitive comparisons, we select the solution closest to the origin in the Pareto solution set to be

**Table 7. Optional machines for related processes.**

|  | Process name | | |
|---|---|---|---|
|  | **Top and ear plate welding** | **Internal welding** | **External welding** |
| Machine type | Welding Robot 1 | Welding Robot 2 | Welding Robot 3 |
| Optional machines | $[M_1\text{-}M_3]$ | $[M_4\text{-}M_5]$ | $[M_6\text{-}M_8]$ |

**Table 8. The processing time and loading and unloading time of the job.**

| Workpiece Type | Workpiece | Processing time | | | | | | | |
|---|---|---|---|---|---|---|---|---|---|
| | | $O_1$ | | $O_2$ | | $O_3$ | | $O_4$* | |
| | | PT | LUT | PT | LUT | PT | LUT | PT | LUT |
| 5tA5A6 | $N_1$ | [37,45,32] | [10,12,12] | [22,25] | [7,8] | [33,33,52] | [12,9,9] | - | [5] |
| 5tA5A6 | $N_2$ | [49,30,43] | [8,10,9] | [28,26] | [5,4] | [53,59,60] | [5,14,14] | - | [5] |
| 5tA5A6 | $N_3$ | [35,50,31] | [10,10,13] | [31,36] | [5,5] | [39,35,40] | [8,13,5] | - | [5] |
| 5tB7B8 | $N_4$ | [33,41,30] | [10,8,9] | [40,38] | [10,5] | [40,35,37] | [13,10,10] | - | [5] |
| 5tB7B8 | $N_5$ | [50,45,43] | [12,10,10] | [21,36] | [2,8] | [44,47,56] | [8,7,15] | - | [5] |
| 5tB7B8 | $N_6$ | [31,48,32] | [12,10,7] | [31,28] | [10,7] | [48,59,45] | [10,5,5] | - | [5] |
| 5tC8C9 | $N_7$ | [45,43,43] | [7,11,13] | [22,30] | [8,7] | [56,46,39] | [12,7,15] | - | [5] |
| 5tC8C9 | $N_8$ | [47,32,36] | [12,13,14] | [35,32] | [9,5] | [40,54,31] | [12,15,10] | - | [5] |
| 5tC8C9 | $N_9$ | [44,47,38] | [9,7,6] | [22,35] | [2,4] | [34,37,40] | [9,15,8] | - | [5] |
| 5tC8C9 | $N_{10}$ | [42,49,47] | [8,8,10] | [25,24] | [5,2] | [54,30,43] | [10,6,9] | - | [5] |

*Notes*: $O_4$* *represents the virtual process of unloading completed workpieces; PT represents processing timing and LUT represents loading and unloading timing.*

recorded. In the solution in this example, the fitness value of the scheduling scheme is [518,309].

Fig 18 depicts a Gantt chart of the solution, including a processing task for each of the processing machines and a transport task for each of the RGVs as well as a loading and unloading task. In Fig 18, different colors represent different workpieces and different colors of the RGVs indicate transport tasks for the corresponding workpieces. The maximum completion time is 518 min, and the constraints of the mathematical model are satisfied.

The optimal values of the trajectories generated by the RGVs are shown in Fig 19. Where the yellow line is the transport generated by the RGV to avoid collision and the green line is the waiting time to avoid collision. There is no overlap during the transport process, which indicates that MOID-GWO effectively plans a conflict-free route.

## 7. Conclusions and future work

In the manufacturing process, CFRP between RGVs increases the complexity of RGVs and production equipment integration scheduling. Therefore, the dual resource integration scheduling of production equipment and RGVs considering CFRP is studied, and the following conclusions are obtained:

**Table 9. Transport time between each machine.**

| From/To | Transport time | | | | | | | | | |
|---|---|---|---|---|---|---|---|---|---|---|
| | $M_0$* | $M_1$ | $M_2$ | $M_3$ | $M_4$ | $M_5$ | $M_6$ | $M_7$ | $M_8$ | $M_9$* |
| $M_0$* | - | 3 | 3 | 6 | 6 | 9 | 9 | 12 | 12 | 15 |
| $M_1$ | 3 | - | 0 | 3 | 3 | 6 | 6 | 9 | 9 | 12 |
| $M_2$ | 3 | 0 | - | 3 | 3 | 6 | 6 | 9 | 9 | 12 |
| $M_3$ | 6 | 3 | 3 | - | 0 | 3 | 3 | 6 | 6 | 9 |
| $M_4$ | 6 | 3 | 3 | 0 | - | 3 | 3 | 6 | 6 | 9 |
| $M_5$ | 9 | 6 | 6 | 3 | 3 | - | 0 | 3 | 3 | 6 |
| $M_6$ | 9 | 6 | 6 | 3 | 3 | 0 | - | 3 | 3 | 6 |
| $M_7$ | 12 | 9 | 9 | 6 | 6 | 3 | 3 | - | 0 | 3 |
| $M_8$ | 12 | 9 | 9 | 6 | 6 | 3 | 3 | 0 | - | 3 |
| $M_9$* | 15 | 12 | 12 | 9 | 9 | 6 | 6 | 3 | 3 | - |

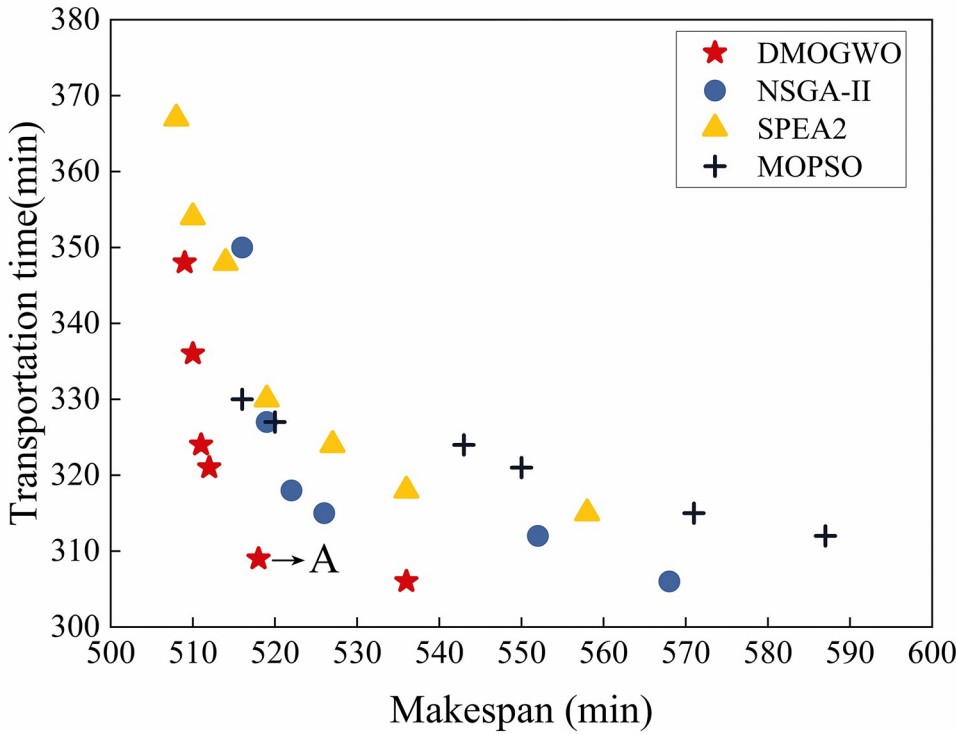

**Fig 17. Pareto front obtained by the proposed MOID-GWO.**

1. DRISP-PERCFR is established. The conflict is divided into loading and unloading conflict, waiting occupation conflict and transportation conflict, and the solution of the conflict with site occupation update is proposed respectively, which improves the accurate mapping degree of the model to the reality.

2. MOID-GWO is proposed for the characteristics of DRISP-PERCFR. MOID-GWO improves the quality and diversity of the initial solution, and solves the dependence degree of the original GWO on the initial solution.

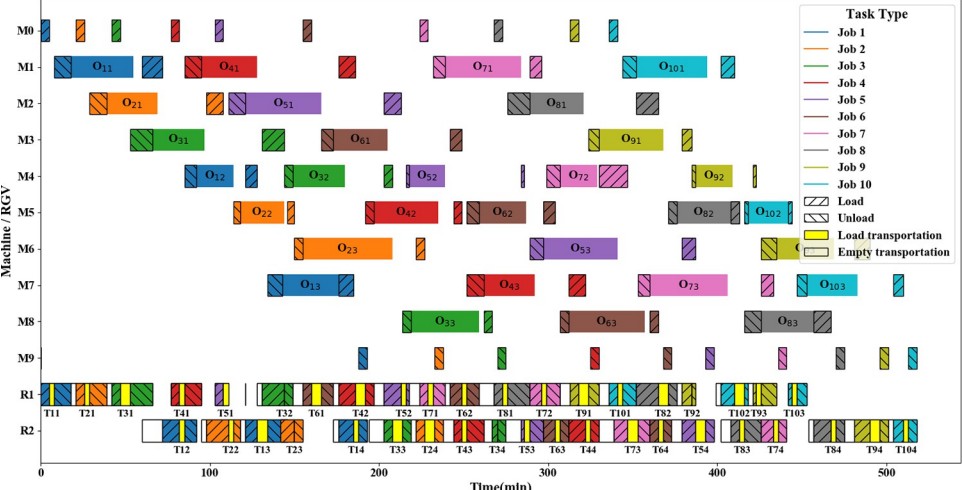

**Fig 18. The Gantt chart of scheduling scheme.**

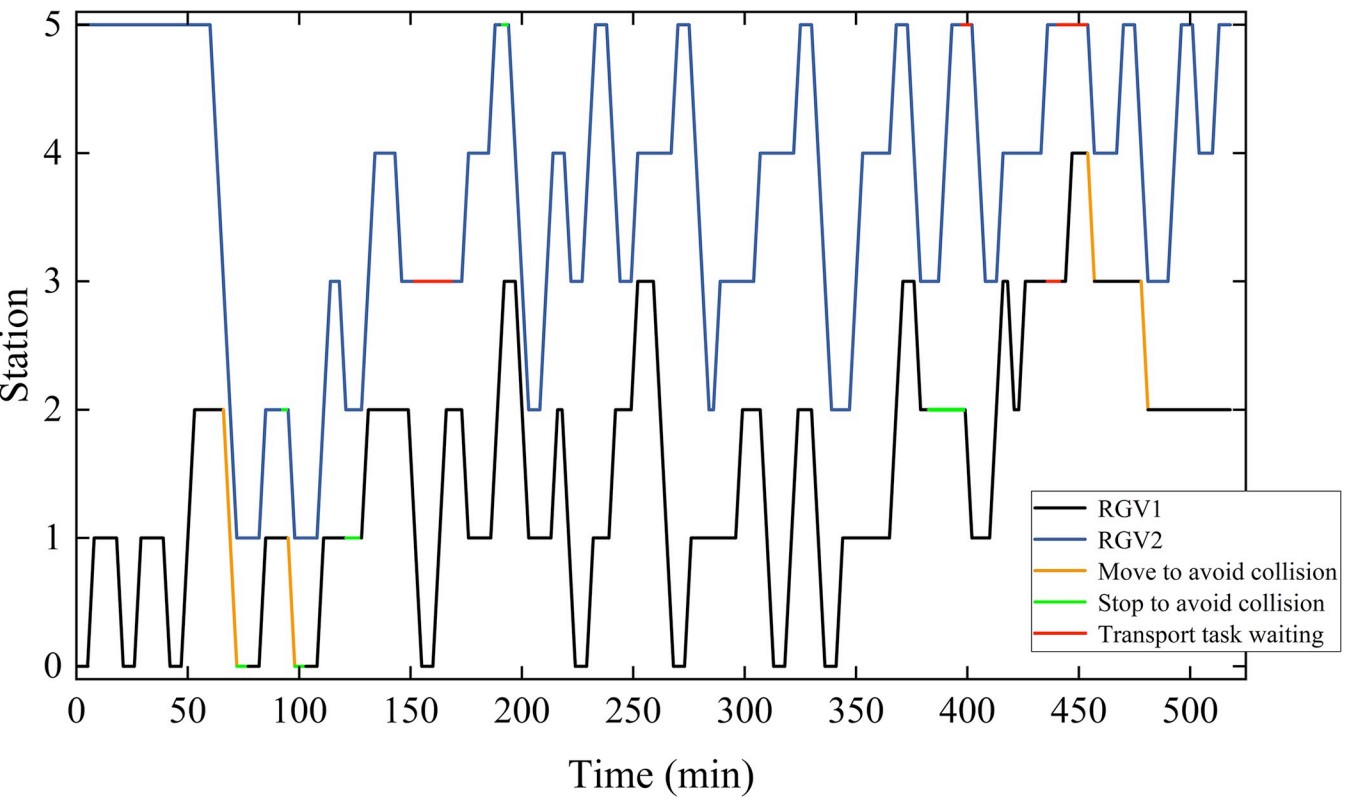

**Fig 19. Trajectories of the RGVs obtained by MOID-GWO.**

3. The feasibility and effectiveness of the proposed model and algorithm are verified in different scale tests and the actual production process of the engineering machinery equipment manufacturing workshop.

Of course, the proposed models and algorithms are limited to some extent. In terms of the model, it is assumed that once the production equipment starts working, the work will not stop. It has weak robustness and anti-interference to the disturbance caused by sudden events. Therefore, dynamic perturbations, such as new workpiece joining, and machine failure, will be considered in DRISP-PERCFR in the future. In terms of the algorithm, although the three-chain encoding of MOID-GWO improves the quality of the solution, it increases the complexity for the subsequent, so new codec forms will be studied in the future.

## Supporting information

**S1 Table. Total distribution of instance data.**
(XLSX)

**S2 Table. Total distribution of instance data.**
(XLSX)

**S3 Table. Optional machines of practical case.**
(XLSX)

**S4 Table. Distribution of practical case.**
(XLSX)

**S5 Table. Transport time between each machine of practical case.**
(XLSX)

## Author Contributions

**Conceptualization:** Qinglei Zhang.

**Data curation:** Zhen Liu.

**Investigation:** Qinglei Zhang, Jianguo Duan.

**Methodology:** Qinglei Zhang, Jing Hu.

**Project administration:** Qinglei Zhang.

**Supervision:** Zhen Liu, Jianguo Duan.

**Validation:** Jing Hu, Zhen Liu.

**Visualization:** Jianguo Duan.

**Writing – original draft:** Qinglei Zhang, Jing Hu.

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
