## [Decision Letter · Decision Letter 0]

6 Nov 2023

PONE-D-23-32956Multi-objective optimization of dual resource integrated scheduling problem of production equipment and RGVs considering conflict-free routingPLOS ONE

Dear Dr. Hu,

Thank you for submitting your manuscript to PLOS ONE. After careful consideration, we feel that it has merit but does not fully meet PLOS ONE’s publication criteria as it currently stands. Therefore, we invite you to submit a revised version of the manuscript that addresses the points raised during the review process.

Please submit your revised manuscript by Dec 21 2023 11:59PM. If you will need more time than this to complete your revisions, please reply to this message or contact the journal office at plosone@plos.org. Please include the following items when submitting your revised manuscript:A rebuttal letter that responds to each point raised by the academic editor and reviewer(s). You should upload this letter as a separate file labeled 'Response to Reviewers'.A marked-up copy of your manuscript that highlights changes made to the original version. You should upload this as a separate file labeled 'Revised Manuscript with Track Changes'.An unmarked version of your revised paper without tracked changes. You should upload this as a separate file labeled 'Manuscript'.

We look forward to receiving your revised manuscript.

Kind regards,

Mazyar Ghadiri Nejad, Ph.D.

Academic Editor

PLOS ONE

Journal Requirements: 

Reviewers' comments:

Reviewer's Responses to Questions

**Comments to the Author**

1. Is the manuscript technically sound, and do the data support the conclusions?

Reviewer #1: Yes

Reviewer #2: Yes

2. Has the statistical analysis been performed appropriately and rigorously? 

Reviewer #1: Yes

Reviewer #2: Yes

3. Have the authors made all data underlying the findings in their manuscript fully available?

Reviewer #1: Yes

Reviewer #2: Yes

4. Is the manuscript presented in an intelligible fashion and written in standard English?

Reviewer #1: Yes

Reviewer #2: Yes

5. Review Comments to the Author

Reviewer #1: Thank you for inviting me as a reviewer for this manuscript “Multi-objective optimization of dual resource integrated scheduling problem of production equipment and RGVs considering conflict-free routing”. The paper is written well and has enough contribution to be published in PLOS ONE. However, hoping to assist the authors in their research efforts, I provide several suggestions for improving the presented work:

- Abstract must be more precise with focus to objective, why the study is needed?

- Brief results must be shown in Abstract and at present it is missing.

- Why you have used Grey Wolf algorithm? Why not other heuristic algorithms like SA, hill-climbing algorithm, GA, ABC, etc.? Discuss advantages of these algorithms.

- You should extend the literature review with application of heuristic algorithms and discuss them to show gap. Remove papers published before 2018. I suggest authors to read and discuss below interesting papers: Das, M., Roy, A., Maity, S., Kar, S., & Sengupta, S. . (2022). Solving fuzzy dynamic ship routing and scheduling problem through new genetic algorithm. Decision Making: Applications in Management and Engineering, 5(2), 329–361.;

Mzili, I., Mzili, T., & Riffi , M. E. (2023). Efficient routing optimization with discrete penguins search algorithm for MTSP. Decision Making: Applications in Management and Engineering, 6(1), 730–743.;

Chaki, S., & Bose, D. . (2022). Optimization of spot-welding process using Taguchi based Cuckoo search algorithm. Decision Making: Applications in Management and Engineering, 5(2), 316–328.

- Provide better validation section with comparisons with the existing models. Add more discussion on the results and discuss advantages and limitations.

- Comparison can be made effective too with more results and discussion and also expand the experimental setup.

- Limitations - Addressing your research limitations could enhance the credibility, applicability, and impact of your research. It is important to note that limitations in a research paper do not necessarily imply negative aspects but rather areas that offer opportunities for further refinement and improvement. Identifying and discussing these limitations transparently can contribute to the overall growth and effectiveness of the study. Explicitly acknowledge the limitations of the proposed framework and model. Address any potential drawbacks or constraints and how they were managed or could be improved in future iterations.

- End the paper with a section on potential future research directions. Highlight areas where further studies could extend or build upon your work.

Reviewer #2: The authors could provide good work. However, there are some concerns to be resolved.

The abstract lacks focus and clarity; it also doesn’t include the motivation for the proposed methods.

The structure of the introduction is not in a standard form. first, we need to find the basics of your problems and then the main aspects and issues related to them. Do not pay attention to lots of details (especially the foundation of the problem)

Improve the literature review. Add several pieces of research in 2019-2023. Moreover, the following references should be used:

A robust possibilistic programming framework for designing an organ transplant supply chain under uncertainty. Annals of Operations Research, 1-38.

Two-echelon electric vehicle routing problem with a developed moth-flame meta-heuristic algorithm. Operations Management Research, 1-22.

Efficient multi-objective meta-heuristic algorithms for energy-aware non-permutation flow-shop scheduling problem. Expert Systems with Applications, 213, 119077.

Designing a portfolio-based closed-loop supply chain network for dairy products with a financial approach: Accelerated Benders decomposition algorithm. Computers & Operations Research, 155, 106244.

Integration of blockchain-enabled closed-loop supply chain and robust product portfolio design. Computers & Industrial Engineering, 179, 109211.

Just-in-time scheduling in identical parallel machine sequence-dependent group scheduling problem. Journal of Industrial & Management Optimization.

Most of the methodological choices lack a clear motivation, and their impact on performance is not analyzed on the manuscript. On the whole, there is no clear indication of where the authors see the main innovation and value of the methodology described.

There is no conceptual comparison with existing approaches, no discussion on the benefits and drawbacks of the new approach. Thus discussions and comparative analyses should be added, also it is important to compare your method with literature ones.

Conclusion is so brief and it did not conduct all aspects of your research well. for example, limitations and future directions are not prepared well.

6. PLOS authors have the option to publish the peer review history of their article (what does this mean?). If published, this will include your full peer review and any attached files.

Reviewer #1: No

Reviewer #2: No

---

## [Author Response · Author response to Decision Letter 0]

18 Dec 2023

We sincerely thank you for your constructive questions and valuable comments.We have made revisions based on your valuable comments, details of which can be found in the document of ''response to reviewers''.

---

## [Decision Letter · Decision Letter 1]

28 Dec 2023

Multi-objective optimization of dual resource integrated scheduling problem of production equipment and RGVs considering conflict-free routing

PONE-D-23-32956R1

Dear Dr. Jing Hu,

We’re pleased to inform you that your manuscript has been judged scientifically suitable for publication and will be formally accepted for publication once it meets all outstanding technical requirements.

Kind regards,

Mazyar Ghadiri Nejad, Ph.D.

Academic Editor

PLOS ONE

Reviewers' comments:

Reviewer's Responses to Questions

**Comments to the Author**

1. If the authors have adequately addressed your comments raised in a previous round of review and you feel that this manuscript is now acceptable for publication, you may indicate that here to bypass the “Comments to the Author” section, enter your conflict of interest statement in the “Confidential to Editor” section, and submit your "Accept" recommendation.

Reviewer #1: All comments have been addressed

Reviewer #2: (No Response)

2. Is the manuscript technically sound, and do the data support the conclusions?

Reviewer #1: Yes

Reviewer #2: Yes

3. Has the statistical analysis been performed appropriately and rigorously? 

Reviewer #1: Yes

Reviewer #2: Yes

4. Have the authors made all data underlying the findings in their manuscript fully available?

Reviewer #1: Yes

Reviewer #2: Yes

5. Is the manuscript presented in an intelligible fashion and written in standard English?

Reviewer #1: Yes

Reviewer #2: Yes

6. Review Comments to the Author

Reviewer #1: The authors have addressed the point of my concern. I am happy with their corrections. Hence, I would like to recommend this manuscript to be published.

Reviewer #2: The paper is improved well and can be published.

7. PLOS authors have the option to publish the peer review history of their article (what does this mean?). If published, this will include your full peer review and any attached files.

Reviewer #1: No

Reviewer #2: **Yes: **Alireza Goli

---

## [Editor Report · Acceptance letter]

16 Jan 2024

PONE-D-23-32956R1 

PLOS ONE

Dear Dr. Hu, 

I'm pleased to inform you that your manuscript has been deemed suitable for publication in PLOS ONE. Congratulations! Your manuscript is now being handed over to our production team.

Kind regards, 

on behalf of

Assoc. Prof. Dr. Mazyar Ghadiri Nejad 

Academic Editor

PLOS ONE